# Quantifying Cloud Adjustments and the Radiative Forcing due to Aerosol-Cloud Interactions in Satellite Observations of Warm Marine Clouds

Alyson Douglas[1] and Tristan L'Ecuyer[1, 2]

[1]University of Wisconsin-Madison 1225 W. Dayton St Madison, WI
[2]Cooperative Institue for Meteorological and Satellite Studies 1225 W. Dayton St Madison, WI

**Correspondence:** Alyson Douglas (ADouglas2@wisc.edu)

**Abstract.** Aerosol-cloud interactions and their resultant forcing remains one of the largest sources of uncertainty of future climate scenarios. The effective radiative forcing due to aerosol-cloud interactions (ERFaci) is a combination of two different effects, how aerosols modify cloud brightness (RFaci, intrinsic) and how cloud extent reacts to aerosol (cloud adjustments CA, extrinsic). Using satellite observations of warm clouds from the NASA A-Train constellation from 2007 to 2010 along with MERRA-2 reanalysis and aerosol from the SPRINTARS model, we evaluate the ERFaci in warm, marine clouds and its components, the $\mathrm{RFaci}_{warm}$ and $\mathrm{CA}_{warm}$, while accounting for the liquid water path and local environment. We estimate the $\mathrm{ERFaci}_{warm}$ to be -0.32 $\pm$0.16 $\mathrm{Wm}^{-2}$. The $\mathrm{RFaci}_{warm}$ dominates the $\mathrm{ERFaci}_{warm}$ contributing 80% (-0.21 $\pm$0.15 $\mathrm{Wm}^{-2}$), while the $\mathrm{CA}_{warm}$ enhances this cooling by 20% (-0.05 $\pm$0.03 $\mathrm{Wm}^{-2}$). Both the $\mathrm{RFaci}_{warm}$ and $\mathrm{CA}_{warm}$ vary in magnitude and sign regionally, and can lead to opposite, negating effects under certain environmental conditions. Without considering the two terms separately, and without constraining cloud-environment interactions, weak regional $\mathrm{ERFaci}_{warm}$ signals may be erroneously attributed to a damped susceptibility to aerosol.

## 1 Introduction

Aerosol-cloud interactions (ACI) and their impact on cloud radiative effects are a vital component of Earth's radiative balance. Warm clouds, in particular, are susceptible to aerosols, and due to their prevalence and role as "Earth's sunblock", these interactions are critical for regulating Earth's surface temperature (Platnick and Twomey, 1994). Aerosols entering a cloud may become cloud condensation nuclei (CCN) initiating a domino effect wherein the cloud's droplet number increases, reducing the mean droplet radius, brightening the cloud's albedo, dampening its ability to precipitate, and, in theory, increasing its lifetime and radiative effect (Twomey, 1977; Albrecht, 1989). However, it remains unknown to what degree aerosols alter warm cloud radiative forcing as models and observations disagree. Global climate models are prone to uncertainty due to their dependence on parameterizations and inability to explicitly represent all scales of ACI, while satellite observations have poor temporal resolution, and natural covariances with the environment may influence warm cloud response to aerosol (Stevens and Feingold, 2009). In order to understand aerosol-cloud interactions and the resulting change in cloud radiative effect, observation-based

methods must address the inherent limitations of satellite observations by creating a framework to resolve the interplay between clouds, the environment, and aerosol-cloud interactions (Seinfeld et al., 2016).

Correctly quantifying the effective radiative forcing due to aerosol-cloud interactions (ERFaci) of warm clouds specifically is important to establish a climate sensitivity and identify cloud feedbacks (Bony and Dufresne, 2005; Rosenfeld, 2006; Boucher et al., 2013). It has been understood since the early 1990s that low, warm clouds play a leading role in determining future warming scenarios (Slingo, 1990). The micro- and macrophysical responses of warm clouds to ACI lead to numerous, poorly understood cloud feedbacks in the Earth system (Gettelman and Sherwood, 2016). Clouds do not exist in isolation (Stephens,

2005). Clouds are part of an interconnected system; changes to one aspect, such as particle size or liquid water content, has a ripple effect to other components of the Earth system. Likewise, clouds can be thought of residing in a "buffered system" where a clouds response to aerosol perturbations can be invigorated or diminished depending on the conditions in which it is initiated (Stevens and Feingold, 2009). These interconnections lead to a range of cloud responses to aerosol that depend on the local meteorology and cloud state (Douglas and L'Ecuyer, 2019). Both the short and long time scales of ACI and their radiative

forcing are affected by the interconnections they exist in, meaning constraining the ERFaci of warm clouds must go beyond a single measure of the ERFaci globally and distinguish the individual components of the ERFaci, the radiative forcing due to aerosol-cloud interactions (RFaci) and cloud adjustments (CA). To account for the challenges in estimating the cloud radiative response to aerosol, we constrain the influences of the local meteorology and cloud state using a method developed in Douglas and L'Ecuyer 2019, hereafter DL19. The ERFaci$_{warm}$ is separated into the RFaci$_{warm}$ and cloud adjustments determined with

constraints on meteorology following DL19 and estimates of each effect are presented to find the relative contributions of the RFaci$_{warm}$ and cloud adjustments to the ERFaci$_{warm}$. The present study expands upon work done in DL19 by specifying what aspects of the cloud lead to changes in the CRE, whether that be the brightness or cloud extent or both, and whether these changes can negate each other, such as when a cloud shrinks but the brightness increases.

      Warm clouds, like marine stratocumulus and trade cumulus, are the prevailing cloud type over the oceans and dominate

aerosol-cloud interactions (Gryspeerdt and Stier, 2012). Marine stratocumulus over the cold upwelling waters, such as off the west coast of Africa, persist for long periods of time in the stable, low marine boundary layers (Wood, 2012). Cumulus form from marine stratocumulus to cumulus transitions and in the equatorial region as trade cumuli (Sandu and Stevens, 2011). Warm clouds sheer abundance and bright albedo make them important to the radiative balance of Earth, and it should be no surprise that warm clouds contribute the largest amount of forcing to the ERFaci (Christensen et al., 2016). Marine stratocumulus have

been the primary focus of aerosol-cloud-radiation interactions due to their sheet-like, "homogeneous" structure, pervasiveness (∼25% of the Earth at any moment), location near anthropogenic continental emissions, and susceptibility to changes in their CCN (Hahn and Warren, 2007; Platnick and Twomey, 1994).

      The warm cloud albedo has the largest response to aerosol compared to mixed phase or ice phase clouds (Christensen et al., 2016). Twomey was the first to hypothesize the high susceptibility of entirely liquid clouds to aerosol using a simple cloud

model; work since then has confirmed this as the basis of RFaci (Twomey, 1977). Observation- and model-based studies focus on the albedo effect because it is a macrophysical manifestation of microphysical processes. An increase in CCN and decrease in mean droplet radius greatly increases the cloud albedo, and, as such, has significant implications for the radiative balance.

The radiative forcing of the albedo effect, or the sudden microphysical response to aerosol loading (RFaci), is dependent on the activation and eventual microphysical initiation of aerosol as cloud droplets, which can be influenced by local dynamics, the stability of the boundary layer, and the initial cloud state (Su et al., 2010). "Model" conditions simulated by Twomey only exist in the most pristine, stable southern oceans (Gryspeerdt et al., 2017; Hamilton et al., 2014). Depending on the region studied, aerosol can increase the cloud albedo as expected, or in certain cases, lead to a dimming effect, such as when aerosol loading reaches a critical point or the local meteorology regulates the sign and/or magnitude of ACI (Gryspeerdt et al., 2019b; Christensen et al., 2014). Studies conflict to what degree the RFaci dominates the ERFaci, in part because the cloud acts as a "buffered system" and mitigates the RFaci depending on the thermodynamic conditions, making the quantification of the RFaci particularly challenging (Goren and Rosenfeld, 2014; Feingold et al., 2016; Stevens and Feingold, 2009).

Efforts to understand the other component of the ERFaci, cloud adjustments, have been similarly clouded in uncertainty. Cloud lifetime and extent are highly susceptible to aerosol (Dagan et al., 2018). Models have shown that aerosol affects the distribution of liquid throughout the cloud and vertical motion within the cloud, greatly perturbing the cloud's lifetime, precipitation, and extent (Ramanathan et al., 2001; Dagan et al., 2016). Aerosol can act to increase the lifetime of clouds through delayed collision coalescence, or decrease the lifetime through evaporation-entrainment and induced cloud feedbacks (Albrecht, 1989; Small et al., 2009). A satellite observation-based study of ship tracks showed clouds experience a expansion or shrinking of cloud extent depending on whether the clouds are at an open or closed state and the background state of the aerosol (Chen et al., 2015). The cloud adjustment response depends on the cloud state and a sequence of reactions dictated by the environment (Gryspeerdt et al., 2019b). As such, cloud adjustments remain the largest source of variability of ERFaci in global climate models (Fiedler et al., 2019).

To account for influences and variation in the $\text{ERFaci}_{warm}$, $\text{RFaci}_{warm}$, and warm cloud adjustments, we constrain the liquid water path, relative humidity of the free atmosphere, and stability of the boundary layer and covariances between them before evaluating the susceptibility of the effect in the same fashion as DL19. These constraints are held fixed first on a global and then on a regional basis to diagnose regime specific then regionally specific responses. Finally, the decomposed $\text{ERFaci}_{warm}$, or the sum of the $\text{RFaci}_{warm}$ and warm cloud adjustments, is calculated, with constraints on the environment and cloud state, for precipitating and non-precipitating scenes on a regional basis. Our methodology aims to reduce biases by accounting for the regionally specific aerosol and thermodynamic conditions (Feingold, 2003). The relationship between aerosol and cloud response has been proven to be sensitive to regional features like aerosol type or meteorology (Twohy et al., 2005; Chen et al., 2014)(DL19). Aerosol-cloud interactions experience a non-linear relationship with liquid water path therefore it is important to separate this complex relationship from ACI and the associated forcing in order to reduce the effects of this non-linear relationship on our results (Gryspeerdt et al., 2019b).

## 2 Methodology and Observations

### 2.1 Data

Collocated satellite observations of cloud shortwave effect, cloud fraction, and aerosol index are obtained by NASA A-Train satellites Aqua, CloudSat, and The Cloud-Aerosol Lidar and Infrared Pathfinder Satellite Observation (CALIPSO) from 2007 to 2010. The NASA A-Train was intentionally created to maximize the synergy between different satellite products in order to improve our understanding of clouds, aerosols, and the environment (L'Ecuyer and Jiang, 2011). Observations of marine warm clouds and aerosols from the Cloud Profiling Radar (CPR) and Moderate Resolution Imaging Spectroradiometer (MODIS)

aboard CloudSat and Aqua, respectively, are utilized to evaluate the effects of aerosol-cloud interactions on the radiative properties of clouds including their albedo and extent.

CloudSat was launched to an orbit collocated with Aqua and other A-Train satellites in 2006. The CPR on CloudSat is a 94 GHz radar with a $\sim$ 1.7 km along track, 1.4 km cross track resolution, and 480 m vertical resolution (Stephens et al., 2018; Tanelli et al., 2008). A number of cloud properties can be inferred using the CPR backscatter including cloud top height, cloud

type, and accompanying radiative effects.

An along track warm cloud fraction is defined using cloud top height from 2B-CLDCLASS-LIDAR and freezing level from 2C-PRECIP-COLUMN. 2B-CLDCLASS-LIDAR combines CloudSat's CPR with CALIPSO lidar observations in order to discern even the thinnest clouds. At each pixel, the cloud fraction is defined by the amount of cloud uptrack and downtrack of that pixel at a 12 km scale, chosen to approximate the scale of marine boundary layer processes and accentuate small scale

changes in extent compared to other large sizes (e.g. 1° x 1°). Using a smaller scale such as 12 kms for cloud fraction will allow even minute changes in the cloud extent to be detected by our methodology; using a larger size such as 96 km ($\sim$1°) may diminish cloud breakup processes within large stratocumulus decks or minimize effects on trade cumuli. 2B-CLDCLASS-LIDAR includes collocated Cloud-Aerosol Lidar with Orthogonal Polarization (CALIPSO) satellite lidar backscatter measurements to identify thin, shallow clouds that may escape detection by the CPR (Sassen et al., 2008). Cloud top heights from

2B-CLDCLASS-LIDAR, defined using a combination of collocated lidar and CPR measurements, are required to be below the freezing level (Haynes et al., 2009). The freezing level of 2C-PRECIP-COLUMN is obtained from European Centre for Medium-Range Weather Forecasts (ECMWF) analyses and is used to separate warm from mixed and ice phase clouds. Focusing only on warm phase clouds helps reduce the uncertainty associated with retrievals of mixed and ice phase clouds.

Cloud fraction is combined with shortwave top of atmosphere forcings from the CloudSat 2B-FLXHR-LIDAR product to

approximate the effect of aerosol on albedo. 2B-FLXHR-LIDAR uses a combination of CPR and CALIPSO measurements along with MODIS cloud properties and atmospheric conditions from ECMWF as input to a radiative transfer model that computes top of atmosphere shortwave fluxes that have been shown to agree well with CERES observations (Henderson et al., 2013). The mean shortwave flux at the top of atmosphere is weighed by a mean incoming solar radiation at the top of atmosphere in our analysis to account for diurnal variation of incoming solar radiation not sampled by the sun-synchronous

A-Train orbit.

We use aerosol index (AI) as a proxy for aerosol concentration from MODIS. The AI is the product of the Angstrom exponent, calculated using aerosol optical depth (AOD) at 550 and 870 nm, and the AOD at 550 nm. AI has been shown to have a higher correlation with CCN compared to AOD (Stier, 2016; Hasekamp et al., 2019). Cloudy scene AI is determined by interpolating between clear scenes along track. This interpolation may reduce the accuracy in completely overcast scenes, however for most scenes where cloud fraction is $< 1$, this interpolation should be sufficiently accurate. Aerosol swelling in high humidity environments also leads to some uncertainty in AI but but should be limited to select high humidity environmental regimes. Pre-industrial aerosol information is provided by Spectral Radiation-Transport Model for Aerosol Species (SPRINT-ARS), an atmosphere-ocean general circulation model (Takemura et al., 2000). Pre-industrial aerosol errors lead to the majority of uncertainty in ACI due to uncertainties in transport, source, and concentration of pre-industrial aerosol conditions (Chen and Penner, 2005).

The sign and regional variations in susceptibilities found using MODIS AI shown within this study were evaluated against susceptibilities found using MACC and SPRINTARS aerosol in order to qualitatively scrutinize any error due to aerosol retrieval (Douglas, 2017). MACC and SPRINTARS provide independent aerosol estimates not susceptible to swelling, instrument sensitivity or retrieval error.. The fact that our results were qualitatively similar using modeled aerosol provides confidence that the derived susceptibilities shown are not simply an artifact of using satellite-derived AI.

The analysis is constrained to clouds with LWPs between 0.02 to 0.4 $\mathrm{kgm}^{-2}$ using the Advanced Microwave Scanning Radiometer for Earth Observing Satellite (AMSR-E), an instrument aboard Aqua that infers water vapor and precipitation amounts using six microwave frequencies over a $\sim 14$ $\mathrm{km}^2$ area (comparable to the averaging scale of our cloud fraction) (Parkinson, 2003; Wentz and Meissner, 2007). While the footprints of CloudSat and AMSR-E do not perfectly overlap, the AMSR-E LWP is used to establish a scene based constraint on the clouds in order to better consolidate our observations into regimes. AMSR-E's footprint is within $\sim 2.5$ km of CloudSat's track, meaning both sensors are observing the same, liquid clouds (Lebsock and Su, 2014). Imposing these LWP limits in place removes only $\sim 1\%$ of observations leaving over 1.8 million satellite observations for analyses, but avoids possible skewing by extremely thick, bright clouds or extremely thin, dim clouds.

Environmental information to define local meteorological regimes is provided by the Modern-Era Retrospective analysis for Research and Applications, version 2 (MERRA-2) reanalysis (Gelaro et al., 2017). To broadly characterize large-scale environmental conditions, MERRA-2 temperature and humidity profiles are collocated by taking the environmental profile within 30 minutes of a CloudSat overpass and within $\sim \frac{1}{2}^{\circ}$ latitude and longitude. Vertical profiles of humidity and temperature are used to calculate the estimated inversion strength (EIS) of the boundary layer and the relative humidity at 700 mb ($RH_{700}$) to represent the humidity of the free atmosphere (Wood and Bretherton, 2006). By simultaneously stratifying the observations by LWP, RH, and EIS, the analysis directly accounts for covariability between LWP and the local environment by separately evaluating the susceptibility of each environmental regime within distinct LWP limits (Douglas and L'Ecuyer, 2019).

Clouds are separated into precipitating and non-precipitating regimes using CloudSat's 2C-PRECIP-COLUMN precipitation flag. Clouds with a 0 precipitation flag, no precipitation detected, are designated as non-precipitating. Precipitating clouds are separated using flag 3, where rain is certain (Haynes et al., 2009). Our precipitating clouds include a majority of the drizzling

cases, as CloudSat's 2C-PRECIP-COLUMN's threshold for drizzle is -15 dB, which should capture all but the lightest drizzling clouds (Stephens and Wood, 2007).

## 2.2 Methodology

In DL19, environmental and cloud state regimes were imposed on a regional basis in order to identify regime specific behavior of aerosol-cloud-radiation interactions. Within each regime, we regressed the cloud radiative effect (CRE) against AI in order to find the susceptibility of warm cloud radiative properties to aerosol. We use these same susceptibilities within section 3.1 to quantify the total warm, marine ERFaci. DL19 found that the susceptibility varies regionally and by regime, however the ERFaci$_{warm}$ depends on the magnitude to which aerosol has increased since pre-industrial times. Further, the ERFaci$_{warm}$ does not diagnose what characteristics of the cloud are causing the effect, prompting us within this paper to decompose the ERFaci$_{warm}$ into the effects on the albedo and the effects on cloud extent.

The mean shortwave flux at the top-of-atmosphere from CloudSat's 2B-FLXHR-LIDAR along with our definition of warm cloud fraction from $60°$ S to $60°$ N are used to define the RFaci$_{warm}$ and cloud adjustment terms of the ERFaci$_{warm}$. We first calculate the ERFaci$_{warm}$ on a regional basis with regime constraints using estimates of the susceptibility of the warm CRE to aerosol from DL19 and pre-industrial and present-day AI from SPRINTARS. We then use a partial derivative decomposition to separate out the RFaci$_{warm}$ and cloud adjustment terms. These terms are evaluated globally as susceptibilities with constraints on the local meteorology and cloud state following the methodology of DL19. The RFaci$_{warm}$ and cloud adjustments are evaluated regionally with constraints on cloud state and local meteorology. The decomposed ERFaci$_{warm}$ is evaluated for precipitating and non-precipitating scenes to account for the potential effects of precipitation on ACI. Finally, the sum of the RFaci$_{warm}$ and cloud adjustments, the decomposed ERFaci$_{warm}$, is compared against the first estimate of the ERFaci$_{warm}$.

## 2.3 Regimes

Following DL19, the ERFaci$_{warm}$ and components are evaluated within a constrained space on both a global and regional scale. LWP is held approximately constant using a set of twelve LWP limits on a global basis and five LWP limits on a regional basis. This is in line with the original work of Twomey, who surmised that only for a fixed LWP will the cloud albedo increase in more polluted conditions. The local meteorology is defined by the stability of the boundary layer and the relative humidity of the free atmosphere. Both the stability, characterized by the estimated inversion strength, and the relative humidity of the free atmosphere, defined at the 700 mb level, have been shown to influence the sign and magnitude of the susceptibility of the CRE to aerosol (Wood and Bretherton, 2006; Ackerman et al., 2004; De Roode et al., 2014). The resulting regimes isolate the susceptibility of the cloud to aerosol under controlled conditions. Buffering can entail the cloud being too thick and impervious to changes due to aerosol due to its high LWP, offsetting and opposite reactions of the cloud resulting in reduced mean signal, or the environment acting to damp the cloud reaction, such as an unstable boundary layer reducing the impact of aerosol on cloud lifetime (Fan et al., 2016; Stevens, 2007). Using EIS and RH$_{700}$ does not guarantee to limit all covariability between the environment, aerosols, clouds, and their interactions. Some covariability may still exist, such as surface winds that may affect both clouds and aerosol (Nishant and Sherwood, 2017). These constraints only account for the major environmental controls

on clouds and aerosol-cloud interactions, some more minor or less common environmental controls may still exert an influence on our results.

While binning our observations by environmental regime should control for some modulation the environment has on aerosol-cloud interactions, it does not fully capture aerosol-environment interactions. For example, in some regions such as off the coast of Africa, biomass burning results in smoke layers that absorb incoming radiation and warm the atmosphere (Cochrane et al., 2019). This could affect the humidity and temperature of the local environment. Environmental regime constraints would capture how the altered environment may regulate aerosol-cloud interactions, but separation into such regimes does not address how the aerosol has impacted the environment.

## 2.4  Decomposing the ERFaci

A Newtonian-based method is employed to represent the $\text{ERFaci}_{warm}$ as a sum of its parts, the $\text{RFaci}_{warm}$ and cloud adjustments. A positive $\text{ERFaci}_{warm}$, $\text{RFaci}_{warm}$, or cloud adjustment denotes a damped cooling effect of the cloud while a negative sign denotes an additional cooling due to aerosol-cloud interactions. If the shortwave cloud radiative effect is the product of the cloud fraction (CF) and the cloudy sky shortwave flux at the top-of-atmosphere ($\text{SW}_{Cloudy}$):

$$\text{CRE} = \text{CF} \times \text{SW}_{\text{Cloudy}} \tag{1}$$

then, taking the derivative of the CRE with respect to the log of aerosol index, we find the effective radiative forcing due to aerosol-cloud interactions ($\text{ERFaci}_{warm}$) or the change in the CRE with respect to aerosol:

$$\text{ERFaci} = \frac{\partial \text{CRE}}{\partial \ln(\text{AI})} \times \Delta \ln(\text{AI}) \tag{2}$$

where $\Delta \ln(\text{AI})$ is the change in $\ln(\text{AI})$ from pre-industrial to present-day conditions derived from SPRINTARS. SPRINTARS is a 3-D aerosol model that includes emission, advection, diffusion, chemistry, wet deposition, and gravitational settling of multiple species of aerosol driven by a general circulation model developed by the University of Tokyo (Takemura et al., 2000, 2005). The results shown herein depend on the emissions scheme from SPRINTARS; if the model were altered, it is possible the total forcing would change due to different $\Delta \ln(\text{AI})$.

All susceptibilities are found using MODIS AI, while only the $\Delta \ln(\text{AI})$ term uses SPRINTARS modeled aerosol. The lowest 12% of aerosol indices are ignored when determining a susceptibility, as these have been shown to have little to no correlation with CCN compared to higher indices (Hasekamp et al., 2019). Error in MODIS AI estimates adds the greatest source of uncertainty in the observationally based portion of this study, however, signals derived are all robust enough to be observed even when random error is added to 10% of the AI estimates. The regressions within all regime constraints, from only meteorological to regional, remain robust for all susceptibilities when 10% of the AI estimates were randomly assigned. The same relationships can be qualitatively observed when SPRINTARS AOD is used in lieu of MODIS AI (Douglas, 2017).

The susceptibility ($\frac{\partial \text{CRE}}{\partial \ln(\text{AI})}$) can be obtained directly from satellite estimates of top-of-atmosphere clear-sky and all-sky fluxes and aerosol index or further decomposed into separate albedo and cloud fraction responses using Equation 1. Applying the chain rule to equation 2, combined with the definition of CRE from Equation 1, gives:

$$\frac{\partial \text{CRE}}{\partial \ln(\text{AI})} = \frac{\partial \text{CF}}{\partial \ln(\text{AI})} \times \overline{\text{SW}}_{\text{Cloudy}} + \overline{\text{CF}} \times \frac{\partial \text{SW}_{\text{Cloudy}}}{\partial \ln(\text{AI})} \tag{3}$$

where the overbars represent means.

The sum of the right hand terms represent the decomposition susceptibility:

$$\text{Decomposition Susceptibility} = \lambda_{Sum} = \frac{\partial \text{CF}}{\partial \ln(\text{AI})} \times \overline{\text{SW}}_{\text{Cloudy}} + \frac{\partial \text{SW}}{\partial \ln(\text{AI})} \times \overline{\text{CF}} \tag{4}$$

The first term of Equation 4 represents the cloud adjustment susceptibility to aerosol, which to first order is the effect of aerosol on the cloud extent:

$$\text{Cloud Adjustment Susceptibility} = \lambda_{CA} = \frac{\partial \text{CF}}{\partial \ln(\text{AI})} \times \overline{\text{SW}}_{\text{Cloudy}} \tag{5}$$

The cloud adjustment forcing is the product of the cloud adjustment susceptibility $\lambda_{CA}$ and the change in AI from pre-industrial to current times $\Delta \ln(\text{AI})$:

$$\text{Cloud Adjustment Forcing} = \lambda_{CA} \times \Delta \ln(\text{AI}) \tag{6}$$

The cloud adjustment susceptibility ($\lambda_{CA}$) is described by its most notable effect, the enhancement and sustainment of clouds as a result of precipitation suppression. We define the cloud adjustments as the product of the change in cloud fraction with respect to aerosol index and the mean cloud shortwave forcing. By multiplying by the mean cloud shortwave forcing, a change in cloud extent is converted to a change in the reflected shortwave. While this term does not explicitly account for precipitation, we separate clouds by rain state and determine the difference in the RFaci$_{warm}$ and cloud adjustments between precipitating/non-precipitating clouds; this difference is likely close to the overall effect of precipitation on aerosol-cloud-radiation interactions.

This cloud adjustment term accounts for the main process, the change in extent of clouds by aerosol, however many other studies define the cloud adjustment term by the change in LWP by aerosol. We choose to instead focus on the expansion or shrinking of clouds by aerosol and constrain any LWP effects. Research has yet to establish how and where LWP increases or decreases due to aerosol-cloud interactions; focusing on the changes to cloud extent reduces the error in the adjustment term due to this uncertainty.

The second term on the right hand side of Equation 4 represents susceptibility of warm cloud radiative forcing due to aerosol-cloud interactions (RFaci):

$$\text{RFaci Susceptibility} = \lambda_{RFaci} = \overline{\text{CF}} \times \frac{\partial \text{SW}_{\text{Cloudy}}}{\partial \ln(\text{AI})} \qquad (7)$$

where the associated forcing is the product of the RFaci$_{warm}$ susceptibility $\lambda_{RFaci}$ and the change in AI from pre-industrial to current times $\Delta\ln(\text{AI})$:

$$\text{Radiative Forcing due to aci} = \lambda_{RFaci} \times \Delta\ln(\text{AI}) \qquad (8)$$

The RFaci$_{warm}$ susceptibility is the change in the shortwave effect owing to changes in cloud droplet radius, an immediate,
fast response. The outgoing shortwave radiation for cloudy scenes depends on the cloud albedo; a brighter, whiter cloud will reflect more incoming solar radiation, increasing SW$_{\text{Cloudy}}$ at the top of the atmosphere. SW$_{\text{Cloudy}}$ is weighted by the annual solar insolation cycle in order to normalize the term and reduce the impact of changes in the incoming solar flux. RFaci$_{warm}$ is weighted by mean cloud fraction since the net effect of brighter clouds depends on how extensive they are.

Finally, to account for the dependence of each susceptibility (RFaci, CA, and total) on the meteorology and cloud state, each
susceptibility ($\lambda$s from above) is evaluated in distinct EIS, RH, and LWP regimes regionally. The product of each susceptibility and $\Delta\ln(\text{AI})$ is the resulting forcing of the aerosol-cloud-radiation interaction:

$$\text{Forcing} = \sum_{l=1}^{\text{N}_{\text{Reg}}} \sum_{k=1}^{\text{N}_{\text{LWP}}} \sum_{j=1}^{\text{N}_{\text{RH}}} \sum_{i=1}^{\text{N}_{\text{EIS}}} (\lambda_{i,j,k,l} \times \text{W}_{i,j,k,l}) \times \Delta(\ln(\text{AI})) \qquad (9)$$

where $\text{W}_{i,j,k,l}$ is the weighting factor, N is the number of limits imposed, and $\lambda$ is the susceptibility being evaluated (ERFaci$_{warm}$, RFaci$_{warm}$, or CA) regionally ($\text{N}_{Reg}$) with constraints on LWP, EIS, and RH$_{700}$. $\text{W}_{i,j,k,l}$ weights the ERFaci$_{warm}$, RFaci$_{warm}$,
and cloud adjustments by the number of observations in each regime and also by the areal size of the region.

Constraints on LWP reduces the secondary effects of aerosol on LWP or LWP on susceptibility, as aerosol can result in thicker clouds and thicker clouds may have a damped reaction to aerosol. Constraining the meteorology separates signals forced by aerosol and the environment (Stevens and Feingold, 2009). On a global scale the approach outlined in DL19 identifies regime specific behavior; when applied on regional scales, the regimes allow a process level understanding of the mean regional
behavior (Mülmenstädt and Feingold, 2018). This approach is optimal for our satellite based observations where larger scale parameters like AOD, AI, and cloud extent are less impacted by retrieval errors than specific properties of the aerosol.

The RFaci$_{warm}$ and cloud adjustment susceptibilities are first understood with limits on the environment and cloud states on a global scale. Their individual forcings, or the product of the susceptibility and $\Delta\ln(\text{AI})$, are then found with constraints on the environment and cloud state regionally and contrasted against initial estimates of the ERFaci$_{warm}$ evaluated under the
same constraints. The susceptibility estimates are not forcings. Forcings are the product of the susceptibilities ($\lambda_{\text{RFaci}}$ or $\lambda_{\text{CA}}$) and the change in the aerosol index from pre-industrial times to current estimates ($\Delta\ln(\text{AI})$). It is possible that even these estimates of forcing are slightly different than the definition of forcing from the IPCC or model based studies which difference

top-of-atmosphere forcings in polluted vs. non-polluted GCM runs (Penner et al., 2011). The sum of these forcings, which we will term the decomposed ERFaci$_{warm}$, is contrasted against the simple expression for ERFaci$_{warm}$ evaluated directly using

Equation 2. By separating out the individual components of the ERFaci$_{warm}$, the physical processes of aerosol-cloud-radiation interactions can be better understood. The difference between the ERFaci$_{warm}$ and the decomposed ERFaci$_{warm}$ represents uncertainty in the linear decomposition owing to covariability, non-linearity, and other effects not quantified by our approach. In reality, there should be a covariability term at the end of Equation 4 to relate how a change in RFaci$_{warm}$ may affect cloud adjustment processes or vice-versa, however a limitation of satellite observations are that they cannot temporally relate events

meaning covariance between the two terms cannot be accurately quantified (Seinfeld et al., 2016). We focus on the main cloud adjustment, the effect of aerosol on the cloud extent/lifetime, however other cloud adjustment effects exist that our simple calculation of a decomposed ERFaci$_{warm}$ misses, such as how precipitation suppression directly leads to changes in cloud extent or how suppression could lead to a later invigorated state of the cloud and faster dissipation.

Precipitation is indicated by the 2C-RAIN-PROFILE rain rate along the entire 12 km track segment (L'Ecuyer and Stephens,

2002). The decomposition susceptibility is found for precipitating and non-precipitating scenes globally using equation 9. Only the decomposition terms are found separately for precipitating and non-precipitating pixels. The CERES footprint is larger than the CloudSat's, meaning while CloudSat could see an entire 12 km along track segment with no rain, the CERES footprint could still contain rain and influence the regression.

Uncertainty in each effect is found first by assuming the uncertainty in the observations lies in the AI, then by assuming a

majority of the overall uncertainty in the ERFaci$_{warm}$ from error in the pre-industrial aerosol concentration estimates (Hamilton et al., 2014). Error is added randomly to AI to find how aerosol swelling or inaccurate retrievals of aerosol near cloud could alter susceptibility estimates. Aerosols swell in the vicinity of clouds, which increases their size and affects the MODIS retrieved AI (Christensen et al., 2017). To assess how significantly this may affect results we have randomly added errors of 10% to our AI estimates and re-derived all signals with all regime constraints. Even with error in AI, the signals within our environmental and

LWP regimes are robust. Uncertainty in the observations is most likely to come from the AI as CloudSat 2B-FLXHR-LIDAR fluxes have been shown to have at most ~10 Wm$^{-2}$ error in shortwave top-of-atmosphere fluxes (Henderson et al., 2013). The error from AI is then combined with randomly adding error to the pre-industrial AI estimates from SPRINTARS to quantify how error in the pre-industrial aerosol may lead to uncertainty in the ERFaci$_{warm}$, RFaci$_{warm}$, and cloud adjustments. Overall, the majority of uncertainty in any ERFaci estimate lies in the pre-industrial aerosol estimate (Chen and Penner, 2005; Carslaw

et al., 2013; Stevens, 2013).

## 3 Results and Discussion

### 3.1 Estimate of the ERFaci

The warm cloud ERFaci, or the effective radiative forcing due to aerosol cloud interactions is -0.32 Wm$^{-2}$ when found with constraints on the liquid water path, stability, and free atmospheric relative humidity applied regionally. As stated before, a

negative ERFaci/RFaci/cloud adjustment denotes additional cooling due to aerosol-cloud interactions. Figure 1 shows each

component of Equation 9 and the resulting regional distribution of the $\text{ERFaci}_{warm}$. The $\text{ERFaci}_{warm}$ is found applying Equation 2 regionally with regime constraints following DL19. This is within the range reported by the fifth IPCC report (-0.05$\text{Wm}^{-2}$ to -0.95$\text{Wm}^{-2}$) but suggests the net cooling effect is toward the lower end of the expected range. Note, however, that this estimate neglects contributions from cold or mixed phased clouds and land regions (Boucher et al., 2013). This first estimate of the $\text{ERFaci}_{warm}$ represents the sum of all effects of aerosol on the warm cloud radiative effect with no distinction between the $\text{RFaci}_{warm}$ and $\text{CA}_{warm}$ and is representative of how aerosol-cloud interactions may be altering the current radiative budget (Carslaw et al., 2013).

As expected, marine stratocumulus decks in the Southeast Pacific and South Atlantic exhibit the largest $\text{ERFaci}_{warm}$, exceeding -3.0 $\text{Wm}^{-2}$ off the coast of Chile. The peak cooling is observed in the southern hemisphere, where the marine stratocumulus cloud decks subsist due to the strong inversions and cool sea surfaces (Wood, 2012). The storm tracks region in the north Atlantic exhibit a slight cooling, as do the marine stratocumulus off the coast of California, however the southern hemisphere dominates the cooling effect. Some regions where dimming occurs are amplified by the change in emissions of the region, such as the Asian coast.

Interestingly, ACI is responsible for a net warming of as much as 0.6 $\text{Wm}^{-2}$ in the tropical Atlantic and Indian oceans. Diagnosing the cause of this warming cannot be done through the $\text{ERFaci}_{warm}$, as it is impossible to accurately attribute it to a reduced albedo or cloud adjustment process. This signature, in particular, motivates decomposing the $\text{ERFaci}_{warm}$ into the $\text{RFaci}_{warm}$ and cloud adjustment components to allow the instantaneous albedo response to be separated from slower cloud processes. The physical processes resulting in a warming differ between the two components as the cloud adjustments are on a macrophysical scale while the $\text{RFaci}_{warm}$ is due to microphysical interactions between aerosol and CCN. The decomposition in Equation 3 allows the specific underlying physical processes responsible for this positive (warming) forcing to be assessed regionally.

The change in aerosol index is most notable off the coast of Asia and along the European coasts. Emissions from large coastal cities lead to large increases in AI, particularly changes in sulfuric aerosol (McCoy et al., 2017). The AI may have decreased off the coast of Australia due to the overall aerosol size increasing, which would decrease the Angstrom exponent and therefore AI (Carslaw et al., 2017). The northern hemisphere has had much larger changes in AI since pre-industrial times compared to the southern hemisphere due to the differences in anthropogenic activity between the two hemispheres. While the southern hemisphere has not experienced the same extreme changes in AI as the coast of Asia, the strong susceptibility of these warm clouds to aerosol combined with the local expansive clouds leads to a large cooling signal throughout the southern oceans.

## 3.2 Impact of LWP

Cloud LWP plays an integral role in modulating the strength of aerosol-cloud interactions. When first theorized by Twomey in 1977, the LWP of the cloud was considered to be constant as the first effect takes place. With this in mind, we first hold the LWP approximately constant and evaluate the decomposition susceptibility, Equation 4, within distinct LWP regimes. While both the $\text{RFaci}_{warm}$ and cloud adjustments are dependent on LWP, they appear to have inverse relationships (Figure 2). $\lambda_{Sum}$ is found

to increase with increasing LWP, reaching a peak susceptibility between 0.06 and 0.15 kgm$^{-2}$ before asymptotically leveling off in the thickest LWP regime between 0.2 to 0.4 kgm$^{-2}$. For the lowest LWPs, the cloud adjustment susceptibility dominates. This reverses in slightly thicker clouds at around 0.08 kgm$^{-2}$. The RFaci$_{warm}$ susceptibility grows to $\sim 20$ Wm$^{-2}$ln(AI)$^{-1}$ after 0.08 kgm$^{-2}$, while the cloud adjustment susceptibility damps and oscillates around 0 after 0.25 kgm$^{-2}$.

Thicker clouds are less susceptible to precipitation suppression, the key process to initiating many of the cloud adjustments

(Sorooshian et al., 2009; Michibata et al., 2016; Fan et al., 2016). This is reflected in the very muted cloud adjustment susceptibility for higher LWPs past $\sim$0.1 kgm$^{-2}$. This inflection point is also where precipitation is more likely to occur in warm clouds and could be a sign of precipitation modulating the effects of aerosol on the cloud fraction (Lebsock et al., 2008; L'Ecuyer et al., 2009; Stevens and Feingold, 2009). An alternative explanation is that thicker clouds with larger LWPs are more likely to precipitate, scavenging aerosol and weakening the susceptibility. Aerosol-cloud-precipitation interactions complicate cloud

adjustment processes in higher LWP clouds; the true susceptibility may be masked by covariance between aerosol and precipitation in these clouds (McCoy et al.). Precipitation would have an instantaneous effect on many cloud adjustment processes as major sink of liquid water within the cloud and therefore dampening process to other possible adjustments. Our framework for the cloud adjustment effect only considers processes which impact, either directly or indirectly, the cloud fraction. At higher LWPs, there are precipitation and other adjustment processes we do not account for that may later on change the radiative

properties of the clouds, such as invigoration increasing the cloud depth and therefore both the longwave and shortwave cloud radiative effect (Rosenfeld et al., 2008; Koren et al., 2014).

Figure 2 confirms that LWP is intrinsically tied to the cloud albedo and extent necessitating the use of cloud state constraints on the decomposed ERFaci$_{warm}$. While a change in LWP is itself considered a cloud adjustment, it is harder to establish a causal relationship between LWP and aerosol than cloud extent and aerosol due to the manifold of environmental parameters

LWP depends on. LWP being held approximately constant in some subsequent analysis should therefore reduce the impact of the LWP adjustment on cloud extent. While LWP being held approximately constant accounts for some variability in the meteorology, explicitly holding the stability and free atmospheric contributions fixed within regimes of EIS and RH$_{700}$ will further control modulation of $\lambda$ by the environment. Modulation by the environment can include the amplification of the reaction through a stable environment further prolonging the cloud lifetime and therefore extent.

While regime constraints on LWP do reduce the covariability between aerosol-cloud interactions and the role LWP plays in regulating these interactions, it does not remove all sources of covariability between LWP, aerosol, the environment, and cloud properties. Aerosol has been shown to negatively correlate with LWP (Gryspeerdt et al., 2019a). It is possible that this relationship, and the inherent relationship between the environment and LWP, could affect results shown.

### 3.3 Constrained by local meteorology

When further separated by meteorological regimes defined by stability and RH$_{700}$ of the free atmosphere, the influence of the environment becomes clearer as strong variations in both the sign and magnitude of RFaci$_{warm}$ and CA$_{warm}$ with environmental regime are evident (Figure 3). Both the RFaci$_{warm}$ and cloud adjustment susceptibilities show warming responses in the most unstable, driest regimes. This is likely due to both the albedo and cloud extent being heavily influenced by entrainment-

evaporation feedbacks (Small et al., 2009; Christensen et al., 2014). $\lambda_{CA}$ shows a warming in the highest humidity, most stable regimes which may reflect cloud breakup processes like the stratocumulus to cumulus transition.

The total decomposed ERFaci$_{warm}$ susceptibility, given by the sum of both the RFaci$_{warm}$ and cloud adjustments within each individual stability and humidity regime, exhibits strong regime specific susceptibilities demonstrating the importance of understanding the total warm cloud radiative response to aerosol with consideration of the environment. Constraints on meteorology allow us consider how meteorology influences the cloud response to aerosol. Without these constraints, any derived susceptibilities could be attributed environmental responses. While cloud darkening occurs in only the most unstable regime ( $<$ -1.8 K), $\lambda_{CA}$ continues to show a warming response in moderately neutral environments ($\sim$2K). This suggests that the instantaneous response (RFaci) is more sensitive to local meteorology than the slower cloud adjustments.

The dominant cooling of $\lambda_{RFaci}$ and $\lambda_{CA}$ in stable regimes illustrates the potential of a stable inversion to strengthen ACI. The peak cooling of $\lambda_{CA}$ occurs in a relatively dry atmosphere $\sim$27% RH$_{700}$. In this environment, the cloud extent rapidly increases as a response to aerosol, however the cloud is topped by a strong, stable inversion that prohibits much of an deepening of the cloud perhaps instigating the effect to push horizontally rather than vertically (Christensen and Stephens, 2011). $\lambda_{RFaci}$ peaks in stable, but comparatively more moist environments where entrainment of moist air from the free atmosphere promotes activation of all available aerosol to CCN, rapidly increasing the albedo. This response may be similar to other regions where trade cumuli form and the FA is relatively moist (Koren et al., 2014).

Finally, while $\lambda_{RFaci}$ shows less variation in sign, it exhibits more variation in magnitude between meteorological regimes indicating the importance of accounting for meteorological influences in order to capture this specific environmental regime dependence. It is possible with additional constraints, understanding how other components of the meteorology is affecting these terms would become more clear. It is also possible $\lambda_{RFaci}$ is impacted by some semi-direct effects by smoke aerosol which would lead to a cloud dimming and positive susceptibility. Semi-direct effects are not accounted for by our methodology, however aerosol within the cloud layer could lead to cloud breakup processes, a dimmer albedo, and changes to the local environment by the absorbing aerosol.

### 3.4   Constraints on cloud state and local meteorology

As seen in Figures 2 and 3, the susceptibility of each component of the ERFaci$_{warm}$ varies with both cloud state and environmental regime. Therefore, when calculating each component of the ERFaci$_{warm}$, both the meteorology and LWP must be accounted for. To accomplish this, the RFaci$_{warm}$ and CA$_{warm}$ susceptibilities are found with constraints on both the LWP and environment (Figure 4). The shaded region of Figure 4 delineates the 10 to 90% range within each of the 11 cloud states of the susceptibility when further separated by the 100 environmental regimes used in Figure 3. Unlike Figures 2 and 3, $\lambda$ is weighted by frequency of occurrence within each environmental state. This illustrates how the magnitude and sign of each susceptibility can vary by environmental regime even when LWP is held approximately constant. The weighted and summed susceptibility is -5.45 Wm$^{-2}$ln(AI)$^{-1}$ with constraints on LWP, stability, and RH$_{700}$ globally. This is slightly smaller than the susceptibility found in DL19, however that susceptibility took into account all changes in warm cloud CRE to aerosol while our decomposition only accounts for the two largest effects, the albedo and cloud extent susceptibilities to aerosol. The lowest

LWP clouds ($\leq 0.1$ kgm$^{-2}$) contribute most to the net susceptibility due to their abundance but also exhibit the widest range in susceptibilities across different meteorological states.

The two components exhibit different behavior in terms of susceptibility to cloud state (defined here by LWP). The cloud adjustment susceptibility is largest for the lowest LWPs, while the RFaci$_{warm}$ susceptibility peaks around 0.06 kgm$^{-2}$ and gradually declines. This may represent a "sweet spot" of cloud albedo susceptibility. Up to 0.1 kgm$^{-2}$, aerosol are easily activated and there are few processes beyond entrainment and activation to reduce the concentration within the cloud layer. Beyond 0.1 kgm$^{-2}$, where the RFaci$_{warm}$ begins to decrease, the cloud may be influenced by precipitation formation, reducing
the $\lambda_{\mathrm{RFaci}}$ within each environmental regime.

$\lambda_{\mathrm{CA}}$ decreases in magnitude with LWP. Higher LWP clouds, independent of the environment, may be less susceptible to lifetime effects, as was seen in Figure 2. Precipitation suppression, the main driver of cloud adjustments, becomes less likely as LWP increases (Fan et al., 2016; Sorooshian et al., 2009). The thinnest and smallest clouds may have the the largest potential to experience a enhancement effect.

## 3.5   Impact of precipitation and environment on susceptibility


Precipitation formation within the cloud and the environment surrounding modulate the susceptibility. When weighted by the relative frequency of occurrence, rather than overall frequency of occurrence, the susceptibility of precipitating clouds is shown to be much higher in some environments than non-precipitating clouds (Figure 5). Precipitating clouds in humid environments especially, defined as having a RH$_{700} > 44\%$, have a much greater susceptibility than any other regime of clouds.
Unstable clouds show a reduced susceptibility in all cases, with precipitating clouds showing a warming effect in these environments while non-precipitating clouds experience an extremely damped cooling effect. Unsurprisingly, in dry environments and stable environments, precipitation does less to magnify the susceptibility and the difference between precipitating and non-precipitating susceptibilities is reduced.

Precipitating clouds reduce the amount of aerosol available to interact with warm clouds through wet scavenging, yet still
may induce several other processes within the cloud that stimulate a response Gryspeerdt et al.. These include stabilizing the boundary layer through virga, increasing the EIS and therefore susceptibility (Figure 3). Precipitation formation within the cloud induces vertical motion and mixing of the cloud layer, increasing turbulence and mixing of the layer which may increase activation of aerosol and therefore the response of the cloud. Further work must be done to resolve how and to what magnitude precipitation alters the warm cloud radiative susceptibility to aerosol.

## 3.6   Contribution of RFaci and cloud adjustments to global ERFaci


With these considerations in mind, the sum of the RFaci$_{warm}$ and CA, or the decomposed ERFaci$_{warm}$ as we will refer to it, is -0.26 $\pm$.15 Wm$^{-2}$ found using Equation 9 (Figure 7). The components of the ERFaci$_{warm}$, the RFaci$_{warm}$ and cloud adjustments, are found using Equations 5 and 7 and shown in Figure 6. The ERFaci$_{warm}$ from Figure 1 is slightly larger in magnitude than the decomposed ERFaci$_{warm}$. Overall, their regional variations and magnitudes are extremely similar,
suggesting the linear decomposition captures a majority of the ERFaci$_{warm}$. The southern ocean dominates the decomposed

ERFaci$_{warm}$, as is expected based on the susceptibilities. The difference in overall magnitude stems from a stronger dimming effect evaluated in the decomposed ERFaci$_{warm}$ (Figure 6). In the decomposed ERFaci$_{warm}$, more regions experience a decrease in CRE with increasing AI compared to the ERFaci$_{warm}$. This may be due to the definition of the decomposed ERFaci$_{warm}$ that allows either the RFaci$_{warm}$ or CA$_{warm}$ to reduce cooling.

A reduced albedo, or positive RFaci, has been noted by other observation based studies before (Chen et al., 2012). A positive RFaci$_{warm}$ can be caused by multiple processes. A semi-direct effect, where non-activated aerosol acts to decrease the total albedo of the cloud in the case of smoke, reducing the CRE of the cloud and therefore the RFaci$_{warm}$ (Johnson et al., 2004). A decrease in the RFaci$_{warm}$ may also be due to any changes to the distribution of liquid water throughout the cloud layer. In certain environmental conditions, an increase in aerosol may lead to sedimentation within the cloud throughout the entrainment

zone, which could decrease the cloud albedo and therefore CRE (Ackerman et al., 2004). If these two effects combined under the "perfect storm" of aerosol and environmental conditions, the RFaci$_{warm}$ would have a large, positive effect.

The cloud adjustment term likewise undergoes a positive, or damped cooling, response in certain regions. A reduced cloud fraction due to aerosol-cloud interactions has been noted before by others (Small et al. (2009), Gryspeerdt et al. (2016)). Chen et al. (2014) noted a decrease in LWP due to an increase in AI in their observationally based study, while other studies have

indicated the LWP response and therefore cloud fraction response can be either positive or negative (Gryspeerdt et al., 2019a). Any process that alters the cloud's liquid water path, such as evaporation-entrainment, may lead to a decrease in cloud fraction given certain environmental conditions.

The discrepancy between the two estimates of ERFaci$_{warm}$ (0.065 Wm$^{-2}$) may be cloud adjustment effects or covariance between RFaci$_{warm}$ and CA$_{warm}$ not captured by the simple regression employed here. The error between the two lies well

within the bounds of error of both estimates ($\pm$.16 and $\pm$.15). While cloud extent changes are the dominant cloud adjustment effect, changes in liquid water path due to precipitation suppression will have an impact on the radiative forcing as well. Future work on understanding and evaluating the ERFaci$_{warm}$ must include other cloud adjustments and explicitly account for covariance between the RFaci$_{warm}$ and cloud adjustments. Although they occur on different time scales, the RFaci$_{warm}$ could be thought of as reactive to cloud adjustments. So while the cloud adjustment process may take hours, an albedo adjustment

occurs simultaneously.

### 3.7    Regional variation due to precipitation

Figure 5 clearly demonstrates that precipitation plays a leading role in modulating the magnitude of aerosol-cloud interactions and their resultant forcing. The contribution of precipitating and non-precipitating clouds to the ERFaci$_{warm}$ is presented in Figure 8. Precipitation has a large impact on both the RFaci$_{warm}$ and warm cloud adjustment processes, indicated by the

difference in global magnitudes between the two ERFaci$_{warm}$ when separated by precipitation (-.21 Wm$^{-2}$) and not separated by precipitation (Figure 7 -.26 Wm$^{-2}$). Precipitating clouds exhibit different microphysical processes and therefore pathways of aerosol-cloud interactions that lead to an increased susceptibility (-43. Wm$^{-2}$ln(AI)$^{-1}$ vs. -30. Wm$^{-2}$ln(AI)$^{-1}$ weighed individually). However, on average only $\sim$30% of warm clouds observed by CloudSat are precipitating, leading to a smaller net contribution to the total ERFaci$_{warm}$ shown in Figure 8. If in future climates, warm clouds rain more frequently, it is possible

that the decomposed ERFaci$_{warm}$ could increase due to precipitating clouds higher susceptibilities, given the environmental conditions (EIS and RH) remain constant.

In regions where trade cumulus are more prevalent and the marine boundary layer is more unstable, precipitation clouds have the capacity to greatly decrease the cooling due to ERFaci$_{warm}$ (Figures 5, 8). However, this positive ERFaci$_{warm}$ is balanced by their expansive cooling throughout the southern ocean. More regions experience a cooling due to ACI when clouds are

precipitating than not precipitating. Further, due to wet scavenging of aerosol, it is possible that precipitating clouds could reduce semi-direct or direct effects and remove aerosol that could otherwise warm the atmosphere. The possible feedbacks or consequences of changes in precipitation require further research, especially since precipitation is heavily controlled by aerosol type as well as concentration.

## 4 Conclusions

The global distribution of the warm, marine cloud ERFaci and its components, the RFaci$_{warm}$ and cloud adjustments, are found with constraints on cloud state and local meteorology following the methodology of DL19. The total effective radiative forcing due to aerosol-cloud interactions is -0.32 $\pm$0.16 Wm$^{-2}$. The radiative forcing due to aerosol-cloud interactions is -0.21 $\pm$0.12 Wm$^{-2}$. The forcing due to cloud adjustments from aerosol-cloud interactions is -0.05 $\pm$0.03 Wm$^{-2}$. In all cases, constraining the environment and cloud state are found to be critical for reducing error in misrepresenting aerosol-environment effects as

aerosol-cloud interactions. Our estimations of the ERFaci$_{warm}$, as a sum and/or single term, agrees with other estimates of the warm cloud ERFaci$_{warm}$ such as Chen et al. who estimated -0.46 Wm$^{-2}$, and with Christensen et al. who estimated -0.36 Wm$^{-2}$. The latter further showed liquid clouds dominate the ERFaci$_{warm}$ over mixed-phase and ice phase aerosol-cloud-radiation interactions. Thus changes in the warm cloud susceptibility to aerosol perturbations could substantially alter the radiative balance due to the warm cloud dominance of the ERFaci$_{warm}$.

Regionally, the ERFaci$_{warm}$ derived from the linear decomposition into RFaci$_{warm}$ and cloud adjustments agrees moderately well with that derived directly from the SW CRE, proving our method of decomposing the ERFaci$_{warm}$ to the first order components does capture the main effects adequately. Globally, the ERFaci$_{warm}$ is dominated by the RFaci$_{warm}$, however the cloud adjustment term is found to contribute $\sim\frac{1}{5}$ of the total forcing. The cloud adjustments vary regionally in sign and magnitude, meaning in some regions the two effects are additive, while in others they may cancel each other out. In the south

Atlantic, both effects lead to a warming, or positive, forcing as clouds both shrink and dim in this region, most likely due to the prevalence of a drier free atmosphere and unstable boundary layer in this region. In the tropical Pacific, clouds dim while the cloud extent swells, leading to an overall muted cooling effect. Regions like this where the two signals have opposing signals are prime examples of why a decomposition of the ERFaci$_{warm}$ into its components is necessary. The muted signal in the tropical Pacific would most likely be attributed to offsetting reactions in the RFaci$_{warm}$ and CA$_{warm}$, as this region shows

a damped signal of ERFaci$_{warm}$, if not for the knowledge that the RFaci$_{warm}$ and CA$_{warm}$ have opposing responses in this region.

It is possible our simple methodology to evaluate cloud adjustments underestimates the possible forcing due to other adjustment processes or the possible covariance with the RFaci$_{warm}$. If the difference between the ERFaci$_{warm}$ and the sum of the RFaci$_{warm}$ and cloud adjustments is assumed to arise from the missing forcing from other adjustments, the total contribution of the CA$_{warm}$ to the ERFaci$_{warm}$ would increase to -0.11 Wm$^{-2}$, or nearly a third, of the -0.32 Wm$^{-2}$. This would be consistent with a recent estimate by Rosenfeld et al. which found the relationship between Nd and cloud fraction, when constrained by LWP, still had a significant signal. Cloud adjustments are found to dominate over the RFaci$_{warm}$ at the lowest liquid water paths. Thus in regions or climate conditions that support enhanced prevalence of thin clouds, the cloud adjustment term would increase at the expense of the RFaci$_{warm}$.

The southern hemisphere dominates the global ERFaci$_{warm}$ due ubiquitous marine stratocumulus in the South Pacific and South Atlantic. The northern hemisphere storm tracks region in the North Atlantic and marine stratocumulus off California exert $\sim \frac{1}{5}$ the magnitude of forcing observed from the southern hemispheres pristine warm clouds. Warm clouds in the southern hemisphere are predisposed for aerosol-cloud-radiation interactions.

Cloud adjustments and RFaci$_{warm}$ varying regionally in sign and magnitudes implies that there are regions and conditions where the two components could effectively cancel each other out, thwarting short term, observation-based attempts at discerning a noticeable change in cloud radiative effects due to aerosol. Moreover, the character of the clouds does not remain constant. Aerosol interactions that result in brighter clouds covering a smaller area, or dimmer clouds covering a larger area, represent important physical responses that may be masked by direct assessment of ERFaci$_{warm}$ from CRE alone. In these regions especially, care should be given to discerning which effect is dominating and to what magnitude.

*Data availability.* All data used is publicly available. Satellite observations are available as stated in the acknowledgements.

*Author contributions.*

*Competing interests.* The authors have no competing interest that are present

*Acknowledgements.* This work was supported by NASA CloudSat/CALIPSO Science Team grant NNX13AQ32G and through the Jet Propulsion Laboratory, California Institute of Technology CloudSat grant G-3969-1. Special thanks to Toshihiko Takemura at Kyushu University for the SPRINTARS modeled AI. All CloudSat data is available from the CloudSat Data Processing Center (www.cloudsat.cira.colostate.edu). MODIS, AMSR-E, and MERRA-2 Reanalysis data is available from the NASA Earth Data repository (www.earthdata.nasa.gov). SPRINTARS aerosol information is available at the SPRINTARS archive (https://sprintars.riam.kyushu-u.ac.jp/).

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

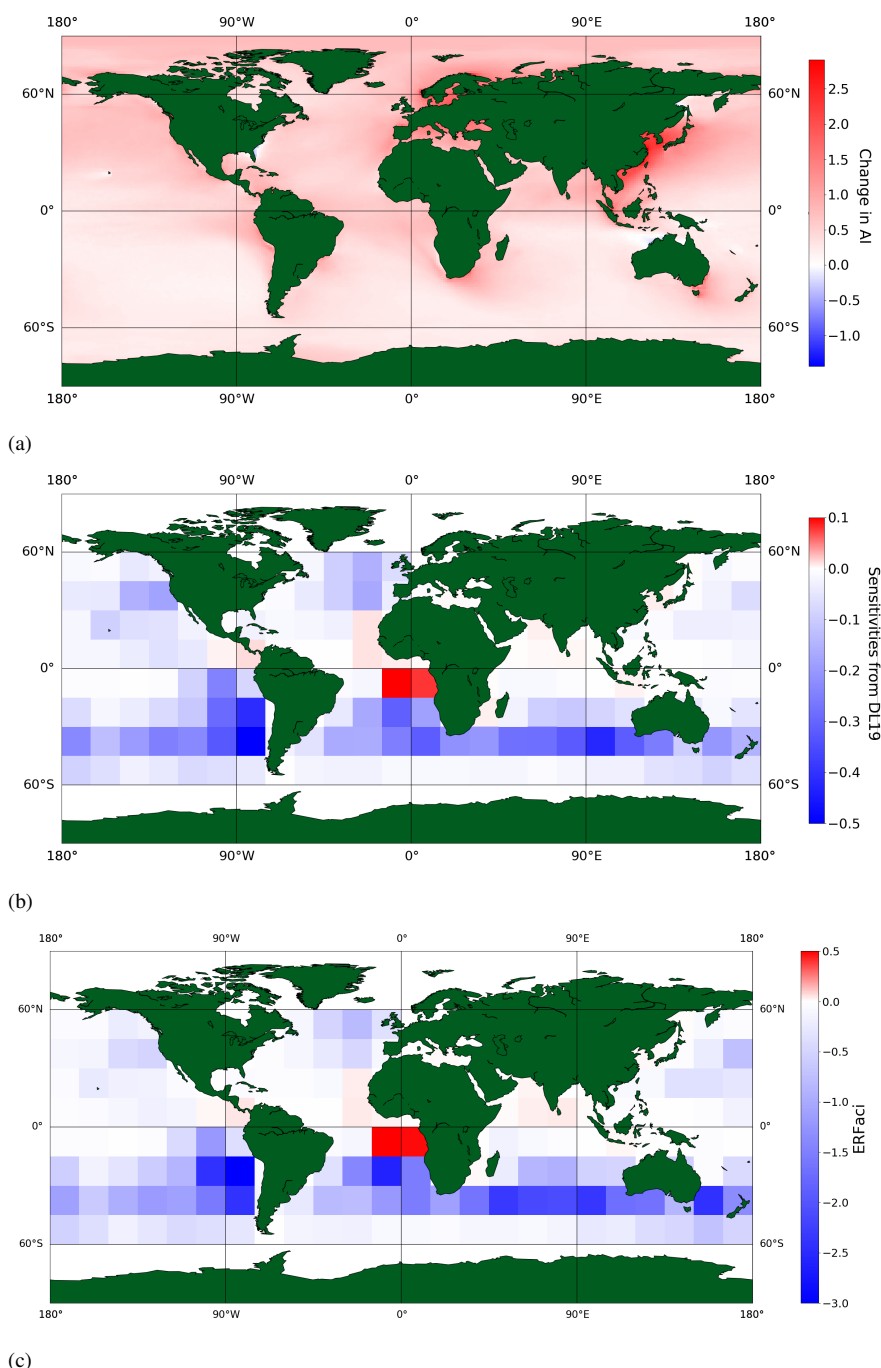

(a)

(b)

(c)

**Figure 1.** The change in aerosol index from SPRINTARS from the pre-industrial to present day (a), $\frac{\partial \text{CRE}}{\partial \ln(\text{AI})}$ adapted from DL19 (b), and the associated $\text{ERFaci}_{warm}$ found using Equation 2 found with constraints on LWP, EIS, and $\text{RH}_{700}$ (c, -0.32 $\pm$.16 $\text{Wm}^{-2}$) using susceptibilities from DL19 (b) without areal weighting.

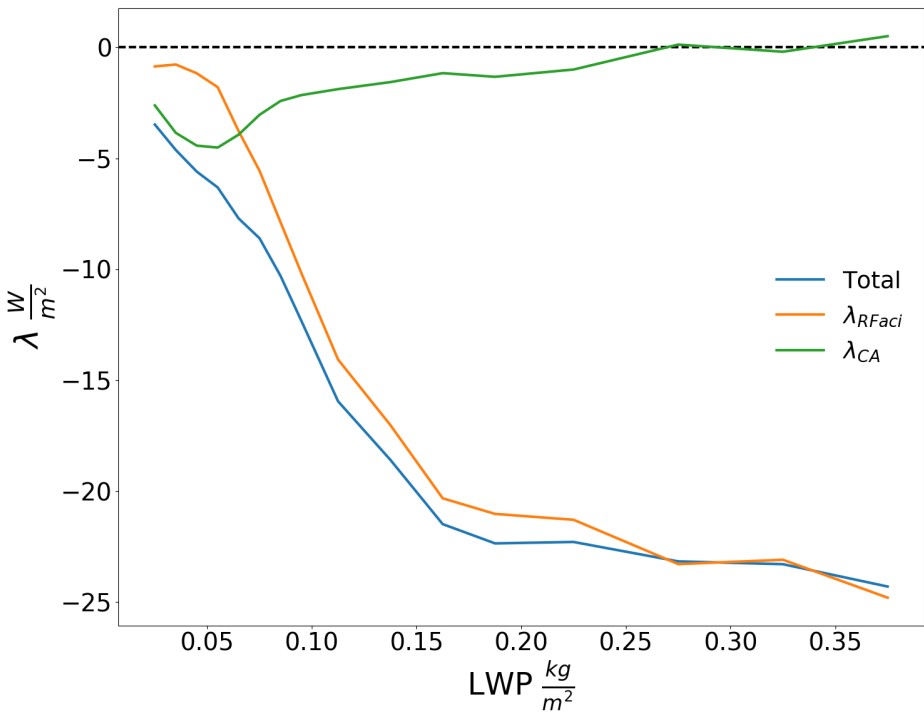

**Figure 2.** The RFaci$_{warm}$, cloud adjustment, and sum of the two susceptibilities, decomposition susceptibility, found within regimes of cloud state defined by LWP. The total decomposition susceptibility is -7.04 Wm$^{-2}$ln(AI)$^{-1}$.

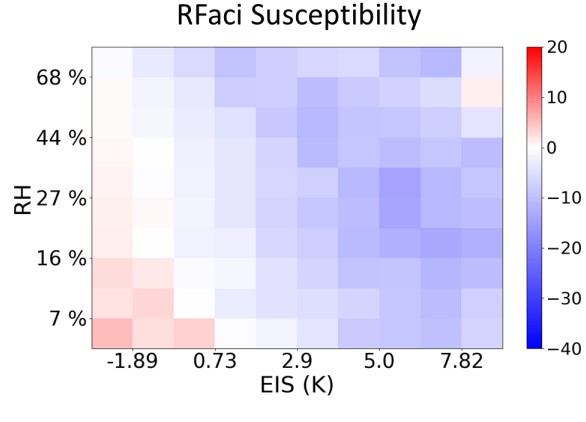

(a)

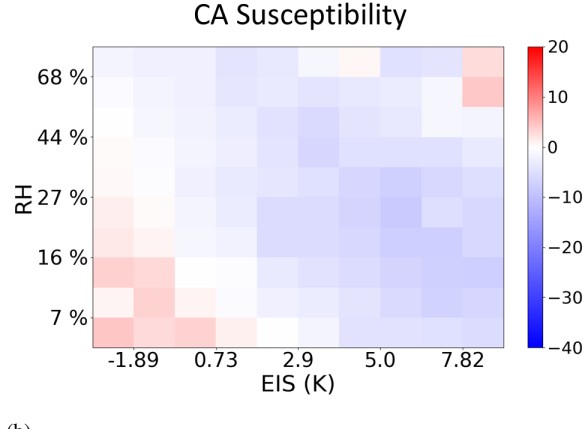

(b)

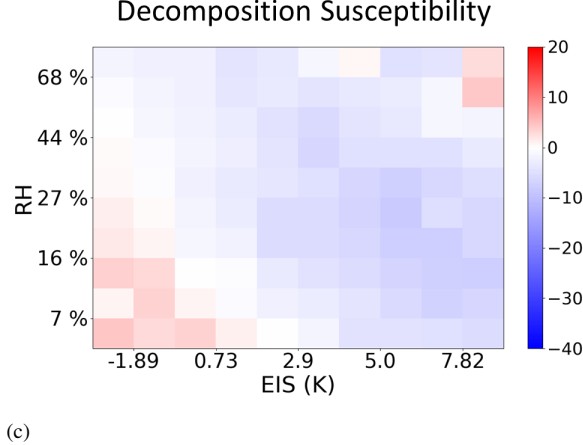

(c)

**Figure 3.** Variations in the a) RFaci$_{warm}$ susceptibility (-5.26 Wm$^{-2}$ln(AI)$^{-1}$), b) cloud adjustment susceptibility (-2.88 Wm$^{-2}$ln(AI)$^{-1}$), and c) the sum of the two susceptibilities, the decomposed ERFaci$_{warm}$ susceptibility (-8.22 Wm$^{-2}$ln(AI)$^{-1}$) with meteorological regimes defined by EIS and RH$_{700}$.

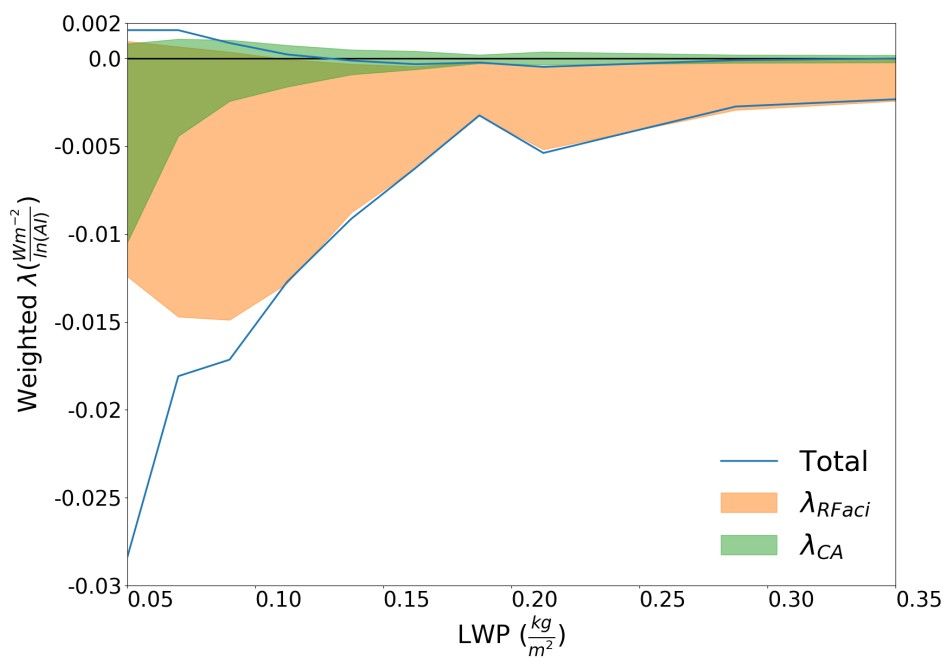

**Figure 4.** 10 to 90% range of the decomposition for 11 cloud states when found within 100 environmental regimes of EIS and $RH_{700}$. The RFaci$_{warm}$ (orange fill, $\lambda_{RFaci}$) and cloud adjustment susceptibilities (green fill, $\lambda_{CA}$) total -4.18 Wm$^{-2}$ln(AI)$^{-1}$ and -1.26 Wm$^{-2}$ln(AI)$^{-1}$, respectively. The sum of the two from 10 to 90 percentiles, the decomposed susceptibility (blue line), totals -5.45 Wm$^{-2}$ln(AI)$^{-1}$.

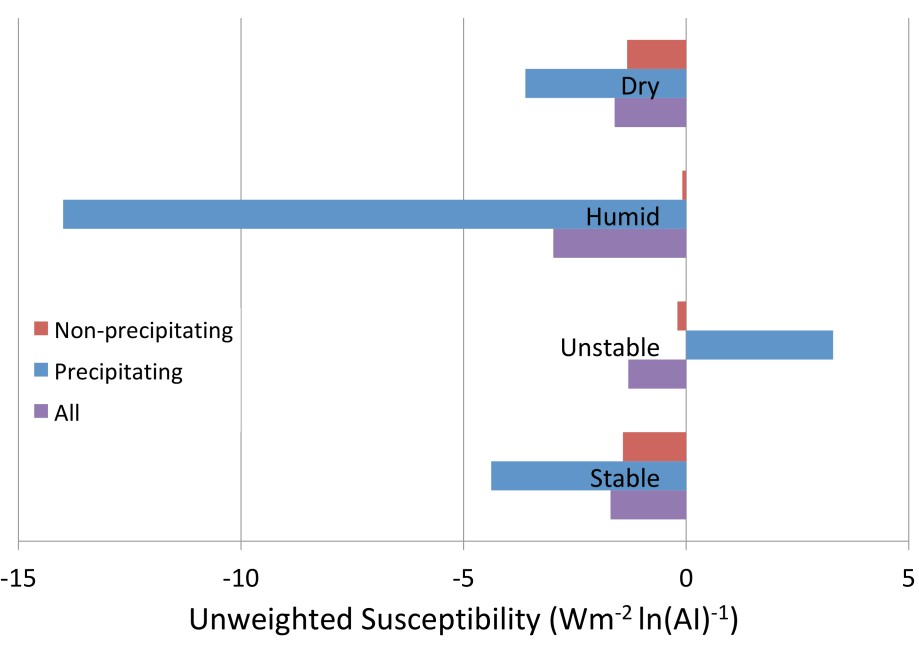

**Figure 5.** Globally summed and relatively weighted susceptibilities for different conditions when found within regimes of EIS, RH, and LWP on a regional basis.

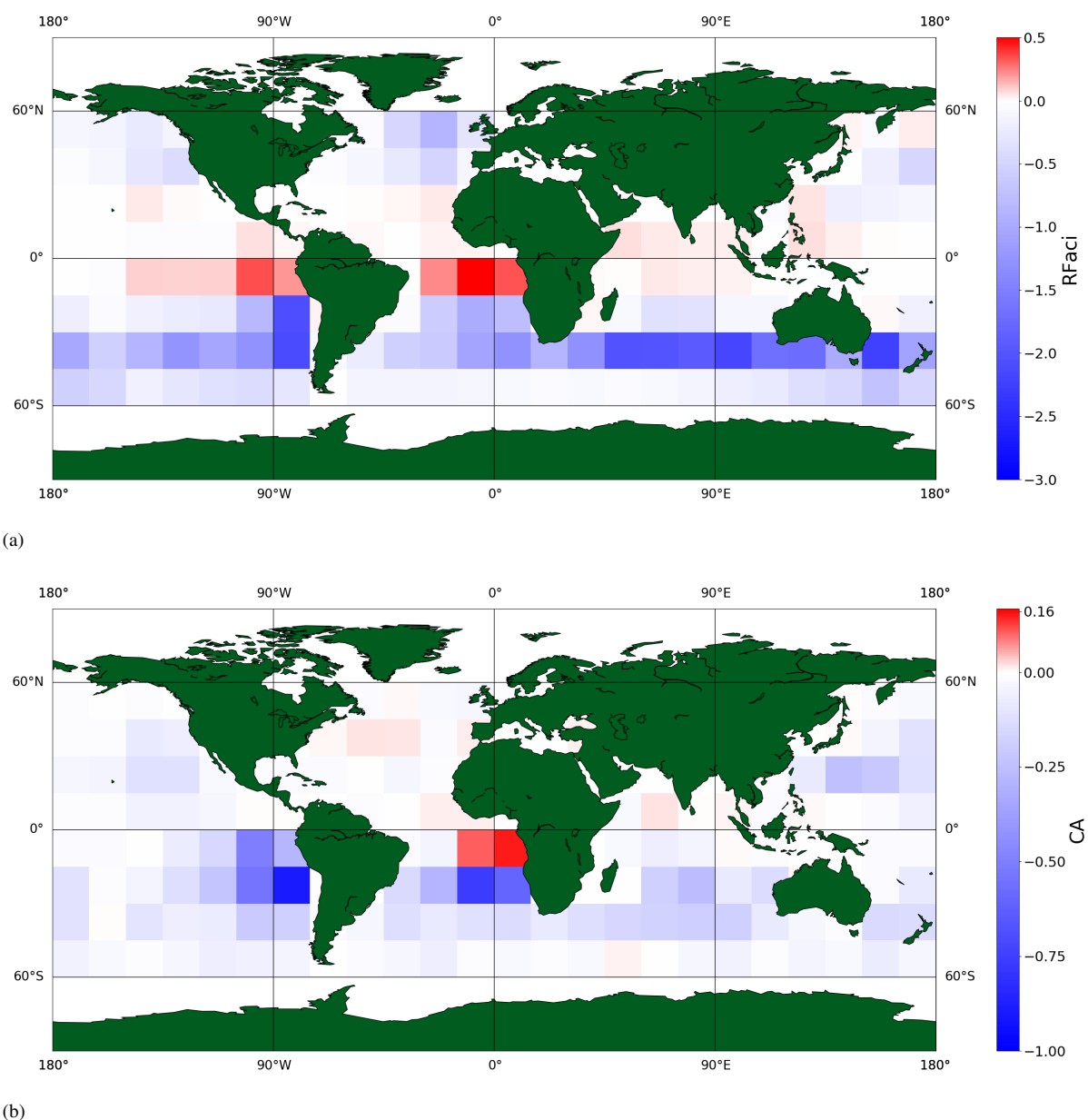

(a)

(b)

**Figure 6.** The radiative forcing due to aerosol-cloud interactions (RFaci) (top, -0.21 $\pm$.12 Wm$^{-2}$) and cloud adjustments (bottom, -0.05 $\pm$.03 Wm$^{-2}$) found on a regional basis with constraints on LWP, EIS, and RH$_{700}$ without weighting by area. Note the colorbar for CA$_{warm}$ (bottom) is 1/3 of the magnitude of RFaci$_{warm}$ (top).

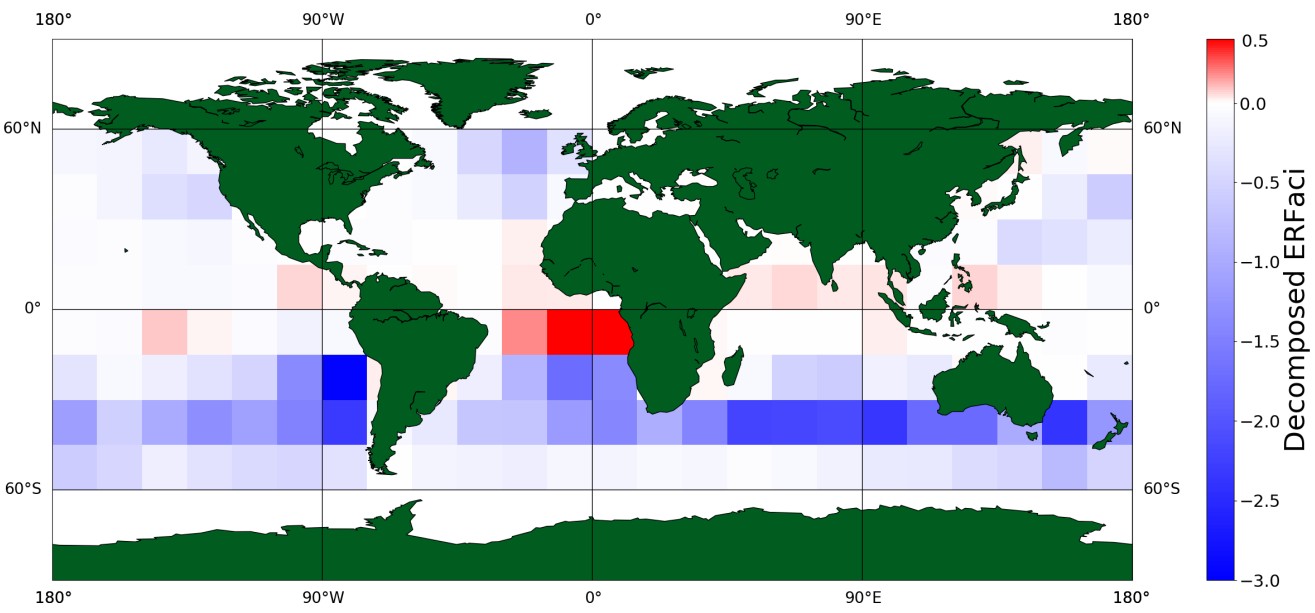

**Figure 7.** The ERFaci$_{warm}$ found as a sum of the RFaci$_{warm}$ and cloud adjustments (Figure 6) with constraints on the LWP, EIS, and RH$_{700}$ on a regional basis (-0.26 Wm$^{-2}$) without areal weighting.

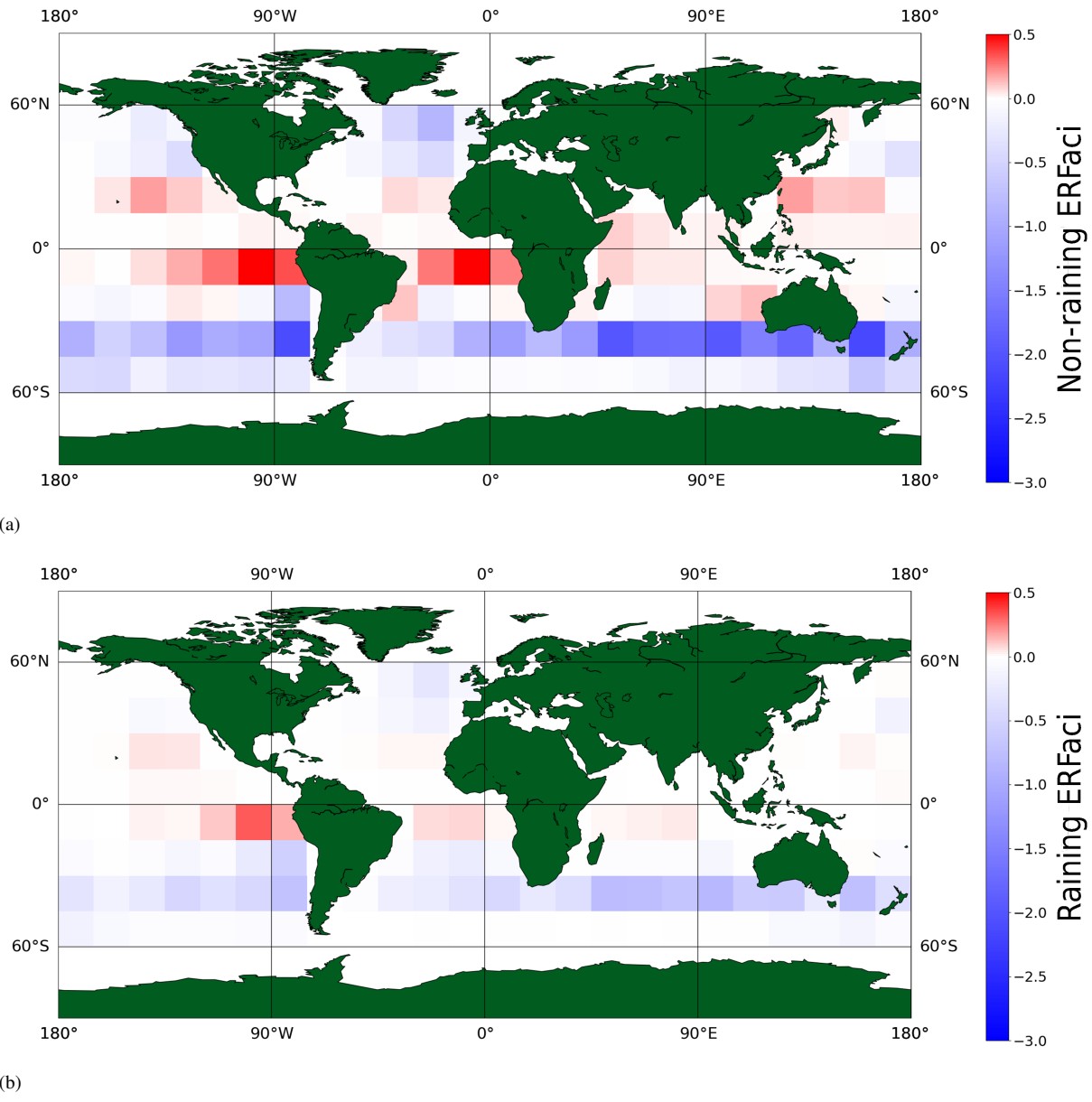

**Figure 8.** The decomposed effective radiative forcing due to aerosol-cloud interactions found as a sum of its components on a regional scale within regimes of EIS, RH, and LWP for a) non-raining clouds (-.147 Wm$^{-2}$) and b) raining clouds (-.06 Wm$^{-2}$).