# Peer review of "Quantifying Cloud Adjustments and the Radiative Forcing due to Aerosol-Cloud Interactions in Satellite Observations of Warm Marine Clouds"

_Atmospheric Chemistry and Physics, 2020_

## Referee Comment (RC1) · Anonymous Referee #1 · 26 Jan 2020

The authors utilize remote sensing observations and a regime-based approach to isolate the effects of varying aerosol index on cloud microphysical (1st indirect effect) and cloud macrophysical properties (adjustments). The authors utilize regimes of above-cloud RH and stability. LWP is binned to account for variations in cloud state in each regime. The results show that in some regions adjustments and the first indirect effect have opposing signs. The authors also show that as LWP increases the radiative response to AI saturates. The analysis presented here satisfies the important problem of separating variability due to meteorology from aerosol-cloud interactions (aci). The

authors find a relatively weak ERFaci from warm-topped clouds over oceans, which appears to be due to dimming in regions in the equatorial Atlantic and Indian ocean.

While I appreciate that the authors are applying the methodology developed in a previous study, it is hard to understand what is being done and I think the authors could briefly summarize their methodology to allow readers to more efficiently refer to DL19. The description of the observational data sets could be much more substantial. It is confusing what observational and modelling data is being used for what. In some cases it appears that observational data sets that are not appropriate are being used, but it is hard to confirm this from the data section. One solution that might make this un-ambiguous would be to create a table of variables and data sources.

A critical issue with this paper is use of area-mean LWP (in-cloud LWP*CF) from microwave when the authors imply they are using in-cloud LWP based on wording in the paper (ln 153). From reading the discussion in DL19 I believe that scene-mean LWP from AMSR is just used to filter data into rough bins, and does not play a role in the analysis beyond this. While this is probably not a big problem, the authors may want to clarify what the footprints of the different data sets are that they are using, possibly with a diagram overlaid over an actual satellite image to allow readers who are less familiar with remote sensing to contextualize what is being shown, especially because the authors are using active instruments averaged along track with passive instruments. In particular, in this regard I am confused how the authors are overlapping along-track averaged CF from Cloudsat-CALIPSO with AMSR LWP and a diagram might be helpful. A nice image of the actual cloud field from MODIS on the background would be helpful to readers trying to contextualize the retrievals in terms of cloud features.

The authors ultimately present a correlative study to predict ERFaci (or at least ERFaci for warm-topped clouds over oceans- see comments below). Characterizing covariance is important but does not guarantee an accurate prediction. In the case of aerosol-cloud adjustments in particular, there is not a unique causality flowing from aerosol to cloud (Wood et al., 2012; Gryspeerdt et al., 2019). In this context, and because their ERFaci is rather weak compared to other studies it seems possible that their analysis conflates aci with precipitation scavenging and other confounders (Gryspeerdt et al., 2019), which would tend reduce correlation strength between aerosol and cloud amount (eg precipitation scavenging is strongest when there is a lot of cloud and there tend to be less cloud and more aerosol off the coast of continents). The authors need to either apply their analysis in a GCM simulating PI and PD (Gryspeerdt et al., 2017; Gryspeerdt et al., 2016; McCoy et al., 2019; Costa-Surós et al., 2019) to make sure that their analysis methodology has predictive power, or examine the response of cloud to some sort of transient change in aerosol (Malavelle et al., 2017; Toll et al., 2019) and make sure that their analysis trained over different data can predict the response to the transient change in AI. Without these falsification tests of their predictions, it is unclear what predictive use their correlation model has in that there is no way to falsify their predictions. Even an approximate calculation using model LWP, CF, SW, and AI without any complex output along the satellite overpass (which doesn't appear to be a major source of error compared to problems from low aerosol amount as shown in Ma et al. (2018)) would provide a much more powerful validation of what the authors are hypothesizing is the ERFaci.

The authors need to refer to their ERFaci as ERFaci_liquid-topped_over_oceans (or at least that is my take from the methodology and Eq 9). Is it possible to use this metric regarding warm-topped maritime clouds, and what they know about the relative occurrence of the clouds that this analysis is performed on, to allow them to extrapolate to global ERFaci? A similar strategy is employed in Costa-Surós et al. (2019) to related forcing over Germany to global forcing.

Specific changes:

Pg 1 ln 3: ERFaci is a combination of microphysical (RFaci) and macrophysical changes (adjustments) and the latter could be further split into changes in extent and thickness(Gryspeerdt et al., 2019; Gryspeerdt et al., 2017; Gryspeerdt et al., 2016). As written this implies that thickness stays constant and the only possible adjustment

is CF. I understand now that this is more like the intrinsic extrinsic separation in other studies (Christensen et al., 2017), but this would be better to clarify in the abstract.

Pg. 2 ln 40: The goals of DL19 overlap a lot with the goals of the present study. A sentence like 'The present study expand on DL19 in the following ways:' would be helpful. I believe the primary difference between these studies is the inclusion of adjustments, but it would be helpful to state that explicitly for readers to rapidly ingest what is happening.

Pg. 3 ln 85: It would help readers to quickly process what data sets are being used to describe what variable to use subheaders here (2.1 Data, 2.1.1 Warm cloud fraction). This is stylistic, but I found it hard to understand where precipitation measurements were coming from. I think that it would help a lot to have a table of what the precise data sets used are, especially since some of the remote sensing data sets being used may be inappropriate, but it is unclear if they are actually used (eg AMSR rain rates, although I believe these are not used despite being mentioned).

Pg4 ln 124: is the material not shown in the citation? If it's in the citation no need to put not shown here.

Pg 4 ln 125: Swelling is a key issue in trying to understand adjustments. I believe that swelling is not an issue for SPRINTARS because the model can be internally consistent, but an additional comment is needed about MACC aerosol swelling. It's unclear that MACC can fully correct for swelling given the very complex way that swelling occurs (Christensen et al., 2017; Twohy et al., 2009). This needs to be explained and caveated. Also, why mix MACC aerosol and MERRA2 meteorology? MERRA2 produces a very similar aerosol reanalysis to MACC and this would avoid confusing MERRA2 meteorology with aerosol reanalysis in a different framework. Also- how are SPRINTARS and MACC not sensitive to precipitation scavenging? Presumably both data sets have a precipitation sink of aerosol otherwise it would be very hard to maintain realistic aerosol.

Pg5 ln126: The MACC AI is effectively satellite AI because it is nudged to satellite radiances. At some point in this paper it is necessary to caveat the use of AI with the results of Ma et al. (2018), which found that satellite inability to detect low aerosol loading biases inferred aci.

Pg5 ln 129: It would be good to note that microwave LWP is area*in cloud LWP. In this context it is a little confusing relating this to Twomey on line 154 because that is for in-cloud LWP, not area mean LWP.

Pg.5 Ln140: This methods section is really short. I understand that the authors refer to DL19, but I think it would help readers evaluate this paper more quickly if a paragraph or so was taken to summarize DL19.

Pg. 5 ln 147: The authors refer to partitioning into precipitating and non-precipitating clouds. I am not clear how this is done. On line 245 it looks like 2C-RAIN-PROFILE is being used- this needs to be caveated that it will only see relatively heavy precipitation, but will miss light rain events. Other parts of the methodology makes it sound like AMSR-E precipitation is being used, which is problematic due to the AMSR-E precipitation just being a partitioning of condensed liquid by SST.

Pg. 6 Eq3-6: how do the authors account for CF being bounded between 0-1 in this calculation?

Pg. 7 Ln 197: The authors assert that they have accounted for the effects of precipitation on the aerosol-cloud-precipitation system. This is not supported by any evidence or citations, but is an important justification of the validity of the analysis presented here.

Pg. 8 ln 221: The authors assert that by binning LWP they reduce the chances of buffering. One thing that should be mentioned in this study is that AI and LWP will naturally anti-correlate due to precipitation and scavenging correlating with cloudiness (eg LWP or CF) (Wood et al., 2012) and due to air mass history leading to both drier

and more aerosol-laden air (Gryspeerdt et al., 2019). These non-causal relationships are not meaningful to ERFaci, but can substantially affect the covariability of cloud macrophysical properties and aerosol, and thus the inferred aci strength (McCoy et al., 2019). It is possible that the LWP binning and precipitation stratification reduce this effect. However, the authors must show some demonstration of the predictive ability of this method by either (1) applying it to GCM data (in this case SPRINTARS) and showing that their methodology when applied in a GCM can accurately reproduce the GCM response to enhance aerosol as in Gryspeerdt et al. (2016) or McCoy et al. (2019) – or – (2) examining one of the transient aerosol emissions identified in recent studies (Malavelle et al., 2017; Toll et al., 2019) and see if their characterization of sensitivity of cloud to aerosol has some predictive ability. Without this sort of test there is no guarantee that the inferred ERFaci_warm-topped_oceanic is accurate.

Pg 9 Ln 261: The authors find an ERFaci that at the very weakest end of what would be expected based on existing best-estimates (Bellouin et al., 2019). This is of course completely fine, but it would be interesting for the authors to add some discussion of why their forcing is so relatively weak compared to other empirical estimates of ERFaci from observations. The authors do cite the AR5 range, but this is for the range of GCM estimates, which may not be as appropriate to consider their results relative to as observational constraint studies. I suspect that this is partially because the authors are not really looking at ERFaci, but ERFaci_warm-topped_oceanic. As such I recommend the authors do not use the terminology ERFaci. In the interest of relating this result to forcing the authors could consider using this methodology applied to GCMs (as noted above I view this as a necessary condition to this analysis) and then using the GCMs to extrapolate this result to ERFaci as in Costa-Surós et al. (2019).

Pg 10 ln 300: An alternative explanation of the weakening precipitation effect in clouds with higher LWP may be that precipitation increases with LWP, which means that precipitation scavenging becomes larger, which in turn means that the true adjustment strength is obscured by non-causal covariance between aerosol and cloud macrophysics (see discussion in McCoy et al. (2019)).

Figure 7 and ln 456: The authors find a large ERFaci in the SH, which is really surprising given the very small change in anthropogenic aerosol in these regions. Figure 1 shows change in AI, but it is a bit hard to distinguish small changes from zero and the authors may want to consider some sort of log normalization to their color scale. However, strong ERFaci exists along a line around 40°S, which is hard to square with studies examining pristine days in the PD (Hamilton et al., 2014). That is to say, the pattern of ERFaci in this study is dramatically different than the RFari shown in, for example, aerocom (Myhre et al., 2013).

Figure 7: While I think it's good to pursue analysis to its conclusion by applying it to all data, I am surprised at the positive RFaci and CA in the tropics. Can the authors comment on whether their analysis is sensitive to retrieval errors in convective cloud? In particular, a positive forcing due to RFaci is quite unusual- while it may be due to biomass burning aerosol above cloud in some regions via semi-direct effects or blocking reflective light (so not really aci) (Bellouin et al., 2019), the appearance of a positive RFaci seems to be more related to SST, than aerosol type given its appearance over the tropics, and far away from strong aerosol sources.

References

Bellouin, N., Quaas, J., Gryspeerdt, E., Kinne, S., Stier, P., Watson-Parris, D., Boucher, O., Carslaw, K. S., Christensen, M., Daniau, A. L., Dufresne, J. L., Feingold, G., Fiedler, S., Forster, P., Gettelman, A., Haywood, J. M., Lohmann, U., Malavelle, F., Mauritsen, T., McCoy, D. T., Myhre, G., Mülmenstädt, J., Neubauer, D., Possner, A., Rugenstein, M., Sato, Y., Schulz, M., Schwartz, S. E., Sourdeval, O., Storelvmo, T., Toll, V., Winker, D., and Stevens, B.: Bounding global aerosol radiative forcing of climate change, Reviews of Geophysics, n/a, 10.1029/2019RG000660, 2019.

Christensen, M. W., Neubauer, D., Poulsen, C. A., Thomas, G. E., McGarragh, G. R., Povey, A. C., Proud, S. R., and Grainger, R. G.: Unveiling aerosol–cloud interactions –

Part 1: Cloud contamination in satellite products enhances the aerosol indirect forcing estimate, Atmos. Chem. Phys., 17, 13151-13164, 10.5194/acp-17-13151-2017, 2017.

Costa-Surós, M., Sourdeval, O., Acquistapace, C., Baars, H., Carbajal Henken, C., Genz, C., Hesemann, J., Jimenez, C., König, M., Kretzschmar, J., Madenach, N., Meyer, C. I., Schrödner, R., Seifert, P., Senf, F., Brueck, M., Cioni, G., Engels, J. F., Fieg, K., Gorges, K., Heinze, R., Siligam, P. K., Burkhardt, U., Crewell, S., Hoose, C., Seifert, A., Tegen, I., and Quaas, J.: Detection and attribution of aerosol-cloud interactions in large-domain large-eddy simulations with ICON, Atmos. Chem. Phys. Discuss., 2019, 1-29, 10.5194/acp-2019-850, 2019.

Gryspeerdt, E., Quaas, J., and Bellouin, N.: Constraining the aerosol influence on cloud fraction, Journal of Geophysical Research: Atmospheres, n/a-n/a, 10.1002/2015JD023744, 2016.

Gryspeerdt, E., Quaas, J., Ferrachat, S., Gettelman, A., Ghan, S., Lohmann, U., Morrison, H., Neubauer, D., Partridge, D. G., Stier, P., Takemura, T., Wang, H., Wang, M., and Zhang, K.: Constraining the instantaneous aerosol influence on cloud albedo, Proceedings of the National Academy of Sciences, 114, 4899-4904, 10.1073/pnas.1617765114, 2017.

Gryspeerdt, E., Goren, T., Sourdeval, O., Quaas, J., Mülmenstädt, J., Dipu, S., Unglaub, C., Gettelman, A., and Christensen, M.: Constraining the aerosol influence on cloud liquid water path, Atmos. Chem. Phys., 19, 5331-5347, 10.5194/acp-19-5331-2019, 2019.

Hamilton, D. S., Lee, L. A., Pringle, K. J., Reddington, C. L., Spracklen, D. V., and Carslaw, K. S.: Occurrence of pristine aerosol environments on a polluted planet, Proceedings of the National Academy of Sciences, 111, 18466-18471, 10.1073/pnas.1415440111, 2014.

Ma, P.-L., Rasch, P. J., Chepfer, H., Winker, D. M., and Ghan, S. J.: Observational constraint on cloud susceptibility weakened by aerosol retrieval limitations, Nature Communications, 9, 2640, 10.1038/s41467-018-05028-4, 2018.

Malavelle, F. F., Haywood, J. M., Jones, A., Gettelman, A., Clarisse, L., Bauduin, S., Allan, R. P., Karset, I. H. H., Kristjánsson, J. E., Oreopoulos, L., Cho, N., Lee, D., Bellouin, N., Boucher, O., Grosvenor, D. P., Carslaw, K. S., Dhomse, S., Mann, G. W., Schmidt, A., Coe, H., Hartley, M. E., Dalvi, M., Hill, A. A., Johnson, B. T., Johnson, C. E., Knight, J. R., O'Connor, F. M., Partridge, D. G., Stier, P., Myhre, G., Platnick, S., Stephens, G. L., Takahashi, H., and Thordarson, T.: Strong constraints on aerosol–cloud interactions from volcanic eruptions, Nature, 546, 485-491, 10.1038/nature22974 http://www.nature.com/nature/journal/v546/n7659/abs/nature22974.html#supplementary-information, 2017.

McCoy, D. T., Field, P., Gordon, H., Elsaesser, G. S., and Grosvenor, D. P.: Untangling causality in midlatitude aerosol-cloud adjustments, Atmos. Chem. Phys. Discuss., 2019, 1-28, 10.5194/acp-2019-665, 2019.

Myhre, G., Samset, B. H., Schulz, M., Balkanski, Y., Bauer, S., Berntsen, T. K., Bian, H., Bellouin, N., Chin, M., Diehl, T., Easter, R. C., Feichter, J., Ghan, S. J., Hauglustaine, D., Iversen, T., Kinne, S., Kirkevåg, A., Lamarque, J. F., Lin, G., Liu, X., Lund, M. T., Luo, G., Ma, X., van Noije, T., Penner, J. E., Rasch, P. J., Ruiz, A., Seland, Ø., Skeie, R. B., Stier, P., Takemura, T., Tsigaridis, K., Wang, P., Wang, Z., Xu, L., Yu, H., Yu, F., Yoon, J. H., Zhang, K., Zhang, H., and Zhou, C.: Radiative forcing of the direct aerosol effect from AeroCom Phase II simulations, Atmos. Chem. Phys., 13, 1853-1877, 10.5194/acp-13-1853-2013, 2013.

Toll, V., Christensen, M., Quaas, J., and Bellouin, N.: Weak average liquid-cloud-water response to anthropogenic aerosols, Nature, 572, 51-55, 10.1038/s41586-019-1423-9, 2019.

Twohy, C. H., Coakley, J. A., and Tahnk, W. R.: Effect of changes in relative humidity on aerosol scattering near clouds, Journal of Geophysical Research: Atmospheres, 114,

n/a-n/a, 10.1029/2008JD010991, 2009.

Wood, R., Leon, D., Lebsock, M., Snider, J., and Clarke, A. D.: Precipitation driving of droplet concentration variability in marine low clouds, Journal of Geophysical Research-Atmospheres, 117, D19210 10.1029/2012jd018305, 2012.

---

## Referee Comment (RC2) · Anonymous Referee #2 · 13 Feb 2020

Review of: Quantifying Cloud Adjustments and the Radiative Forcing due to Aerosol-Cloud Interactions in Satellite Observations of Warm Marine Clouds By Douglas and L'Ecuyer

This study uses satellite observations with the addition of model aerosol data and re-analysis meteorological data to calculate the effective radiative forcing due to cloud-aerosol interaction in warm clouds over the oceans. The authors decompose the forcing to two components: due to the Twomey effect (RFaci), and due to cloud adjustments (CA), which in this case include only changes in cloud cover (without including changes

in LWP). The analysis is conducted also regionally and as a function of LWP, inversion strength and RH in the free troposphere. The binning according to the last two criteria is done to account for the meteorological dependency. The calculation is also done separately for precipitating and non-precipitating clouds. I think that this paper presents some interesting results that worth being published. However, I think that the paper includes some limitations that are not all fully acknowledged in the manuscript. Hence, including a more comprehensive discussion about these limitations and maybe weakening the conclusions accordingly will improve the paper.

General comments

1) If I understand correctly, calculating the radiative forcing based on the multiplication of the susceptibility calculated in present day with the total change in AI between present day and preindustrial assume linearity of the susceptibility with time. As you show that the susceptibility is a function of the environmental conditions and it is known that the environmental conditions changed, it is not clear how valid is this assumption. In addition, I think that your calculation assumes that the frequency of occurrence of each bin of EIS, RH and LWP remain the same between PD and PI (as you do not account for changes in the frequency of occurrence -eq. 9). I can't see any reasons for that to be true. Hence, and because of the large uncertainty in PI aerosol conditions, it might be better to stay only with the susceptibility calculations and not present the forcing calculations. I leave it to the authors to decide. 2) Feedbacks between clouds and the environmental conditions are not discussed and accounted for sufficiently. It is known that the environmental conditions may change differently under different aerosol conditions. In particular, the EIS and RH (maybe not at 700mb but definitely below that) may be affected by the clouds feedback on the environmental conditions differently under different aerosol conditions. In addition, direct aerosol-radiation interaction may influence the environmental conditions. Hence, the binning according to the meteorological conditions may not be independent of the aerosol conditions. I suggest to add a discussion about that. In addition, the separation to precipitating/non-precipitating

conditions could be, under certain conditions, due to aerosol effect (total rain suppression could be found in shallow clouds under polluted conditions). This effect is not discussed and you treat it as if it was external. 3) Co-variability between aerosol and cloudiness and the uncertainty in the causality relationships are not discussed enough. I appreciate that binning the data according to EIS and RH at 700 mb may reduce the role of co-variability between aerosol and cloudiness. However, some co-variability may still remain. For example, it was previously shown (Nishant and Sherwood, 2017) that under some conditions, near surface wind speed have a positive correlation with both aerosol concentration and cloudiness (CF in this case). It is possible, and even expected, that wind speed will be partially corelative with EIS and RH but not sure to what extent. I suggest to add a dissection about those limitations. 4) Uncertainties due to the semi-direct effect are not mentioned. Form satellite observations it is impossible to distinguish between the aerosol microphysical effect and the semi-direct effect but the latter is very likely to affect your calculations. I suggest to include a discussion about that. 5) Referring to the forcing only from warm marine cloud as ERFaci and RFaci might be confusing with the total estimations for all cloud. I appreciate that you mention the focus on warm clouds over the ocean directly in the tile and in many other places but I still think that the use of general terms here could be confusing. 6) At many places along the manuscript you mention "buffering" as if it was an artefact that one should avoid in his analysis (i.e. "While LWP being held approximately constant accounts for some variability in the meteorology, explicitly holding the stability and free atmospheric contributions fixed within regimes of EIS and RH700 will further control buffering and modulation of $\lambda$ by the environment."). I think that if indeed clouds under different aerosol levels change differently the environmental conditions to reduce the total aerosol effect, that is something important to understand. In addition, I think you don't properly define what you mean by "buffering". That term could be used to describe many mechanisms.

Specific comments:

L3: CA is not defined hear. Consider writing in full cloud adjustments.

L10: if RFaci and CA counteract and the total effect is small I would say that it could be attributed to damped susceptibility (or buffering). Why is it "erroneously"?

L21: what do you mean by "cloud forcing"? is it the cloud radiative effect? I think it is better not to use forcing here and stick with the common definition of radiative forcing.

L68-72: consider adding here that the sign of the effect was also shown to be a function of the background aerosol concentration.

L82: the non-monotonic response was shown for other cloud properties (such as cloud fraction and top height) as well as for precipitation. Hence, I don't understand why is it important to separate specifically this effect from the rest.

L105, L109, L141 and other places: again, maybe better to use radiative effect here instead of forcing.

L119: SPRINTARS was run (in the paper you are citing) in a T21 resolution ($\sim$5.6o) and hence is not "cloud resolving" at all.

L250-258: I couldn't really understand how the uncertainty was calculated. I think more details are needed for it to be reproducible. What is the magnitude of error added to PI and PD AI estimations? How did you choose this magnitude?

L265: you are comparing here the estimated forcing for only warm cloud over the ocean with the total estimation from the IPCC report. I don't think this competition is valid. In addition, in the introduction you cited a few papers showing that most of the ERFaci is coming from warm clouds over the ocean. How that can go together with the relatively low estimations you are getting for warm cloud compared to the total forcing?

L310: you cite here a paper focusing on deep convective clouds. Consider adding papers discussing warm cloud invigoration.

L 312: I don't understand the claim here. Why determining the casualty of aerosol

effect on LWP is more difficult than for CF?

L320: why is that a sign of "buffering"? it just means that the aerosol effect is non-monotonic and change sign. The aerosol level at which the sign flip is a function of the environmental condition as was shown before.

L417: the possible change in precipitation could also be relevant between PI and PD making point 1 (general comments above) even more critical.

L426-430: you don't mention here, at the beginning of the conclusion section, that these estimations are only relevant for warm cloud over the oceans. It could look like you are giving general estimations here.

L442: I think that this could also be due to the semi-direct effect of absorbing aerosols.

L445: again, if RFaci and CA counteract and the total effect is small I would say that it could be refer to as "buffering".

Technical comments L102 and L107: ECWMF -> ECMWF? Anyway, should be written in full (and maybe also add a citation).

L401: "on the both the"

Reference Nishant, N., and Sherwood, S. C.: A cloud‐resolving model study of aerosol‐cloud correlation in a pristine maritime environment, Geophysical Research Letters, 44, 5774-5781, 2017.

---

## Author Response (AR1)

The authors utilize remote sensing observations and a regime-based approach to isolate the effects of varying aerosol index on cloud microphysical (1st indirect effect) and cloud macrophysical properties (adjustments). The authors utilize regimes of above-cloud RH and stability. LWP is binned to account for variations in cloud state in each regime. The results show that in some regions adjustments and the first indirect effect have opposing signs. The authors also show that as LWP increases the radiative response to AI saturates. The analysis presented here satisfies the important problem of separating variability due to meteorology from aerosol-cloud interactions (aci). The

authors find a relatively weak ERFaci from warm-topped clouds over oceans, which appears to be due to dimming in regions in the equatorial Atlantic and Indian ocean.

While I appreciate that the authors are applying the methodology developed in a previous study, it is hard to understand what is being done and I think the authors could briefly summarize their methodology to allow readers to more efficiently refer to DL19. The description of the observational data sets could be much more substantial. It is confusing what observational and modelling data is being used for what. In some cases it appears that observational data sets that are not appropriate are being used, but it is hard to confirm this from the data section. One solution that might make this un-ambiguous would be to create a table of variables and data sources.

A critical issue with this paper is use of area-mean LWP (in-cloud LWP*CF) from microwave when the authors imply they are using in-cloud LWP based on wording in the paper (ln 153). From reading the discussion in DL19 I believe that scene-mean LWP from AMSR is just used to filter data into rough bins, and does not play a role in the analysis beyond this. While this is probably not a big problem, the authors may want to clarify what the footprints of the different data sets are that they are using, possibly with a diagram overlaid over an actual satellite image to allow readers who are less familiar with remote sensing to contextualize what is being shown, especially because the authors are using active instruments averaged along track with passive instruments. In particular, in this regard I am confused how the authors are overlapping along-track averaged CF from Cloudsat-CALIPSO with AMSR LWP and a diagram might be helpful. A nice image of the actual cloud field from MODIS on the background would be helpful to readers trying to contextualize the retrievals in terms of cloud features.

The authors ultimately present a correlative study to predict ERFaci (or at least ERFaci for warm-topped clouds over oceans- see comments below). Characterizing covariance is important but does not guarantee an accurate prediction. In the case of aerosol-cloud adjustments in particular, there is not a unique causality flowing from aerosol to cloud (Wood et al., 2012; Gryspeerdt et al., 2019). In this context, and because their ERFaci is rather weak compared to other studies it seems possible that their analysis conflates aci with precipitation scavenging and other confounders (Gryspeerdt et al., 2019), which would tend reduce correlation strength between aerosol and cloud amount (eg precipitation scavenging is strongest when there is a lot of cloud and there tend to be less cloud and more aerosol off the coast of continents). The authors need to either apply their analysis in a GCM simulating PI and PD (Gryspeerdt et al., 2017; Gryspeerdt et al., 2016; McCoy et al., 2019; Costa-Surós et al., 2019) to make sure that their analysis methodology has predictive power, or examine the response of cloud to some sort of transient change in aerosol (Malavelle et al., 2017; Toll et al., 2019) and make sure that their analysis trained over different data can predict the response to the transient change in AI. Without these falsification tests of their predictions, it is unclear what predictive use their correlation model has in that there is no way to falsify their predictions. Even an approximate calculation using model LWP, CF, SW, and AI without any complex output along the satellite overpass (which doesn't appear to be a major source of error compared to problems from low aerosol amount as shown in Ma et al. (2018)) would provide a much more powerful validation of what the authors are hypothesizing is the ERFaci.

The authors need to refer to their ERFaci as ERFaci_liquid-topped_over_oceans (or at least that is my take from the methodology and Eq 9). Is it possible to use this metric regarding warm-topped maritime clouds, and what they know about the relative occurrence of the clouds that this analysis is performed on, to allow them to extrapolate to global ERFaci? A similar strategy is employed in Costa-Surós et al. (2019) to related forcing over Germany to global forcing.

Specific changes:

Pg 1 ln 3: ERFaci is a combination of microphysical (RFaci) and macrophysical changes (adjustments) and the latter could be further split into changes in extent and thickness(Gryspeerdt et al., 2019; Gryspeerdt et al., 2017; Gryspeerdt et al., 2016). As written this implies that thickness stays constant and the only possible adjustment

is CF. I understand now that this is more like the intrinsic extrinsic separation in other studies (Christensen et al., 2017), but this would be better to clarify in the abstract.

Pg. 2 ln 40: The goals of DL19 overlap a lot with the goals of the present study. A sentence like 'The present study expand on DL19 in the following ways:' would be helpful. I believe the primary difference between these studies is the inclusion of adjustments, but it would be helpful to state that explicitly for readers to rapidly ingest what is happening.

Pg. 3 ln 85: It would help readers to quickly process what data sets are being used to describe what variable to use subheaders here (2.1 Data, 2.1.1 Warm cloud fraction). This is stylistic, but I found it hard to understand where precipitation measurements were coming from. I think that it would help a lot to have a table of what the precise data sets used are, especially since some of the remote sensing data sets being used may be inappropriate, but it is unclear if they are actually used (eg AMSR rain rates, although I believe these are not used despite being mentioned).

Pg4 ln 124: is the material not shown in the citation? If it's in the citation no need to put not shown here.

Pg 4 ln 125: Swelling is a key issue in trying to understand adjustments. I believe that swelling is not an issue for SPRINTARS because the model can be internally consistent, but an additional comment is needed about MACC aerosol swelling. It's unclear that MACC can fully correct for swelling given the very complex way that swelling occurs (Christensen et al., 2017; Twohy et al., 2009). This needs to be explained and caveated. Also, why mix MACC aerosol and MERRA2 meteorology? MERRA2 produces a very similar aerosol reanalysis to MACC and this would avoid confusing MERRA2 meteorology with aerosol reanalysis in a different framework. Also- how are SPRINTARS and MACC not sensitive to precipitation scavenging? Presumably both data sets have a precipitation sink of aerosol otherwise it would be very hard to maintain realistic aerosol.

Pg5 ln126: The MACC AI is effectively satellite AI because it is nudged to satellite radiances. At some point in this paper it is necessary to caveat the use of AI with the results of Ma et al. (2018), which found that satellite inability to detect low aerosol loading biases inferred aci.

Pg5 ln 129: It would be good to note that microwave LWP is area*in cloud LWP. In this context it is a little confusing relating this to Twomey on line 154 because that is for in-cloud LWP, not area mean LWP.

Pg.5 Ln140: This methods section is really short. I understand that the authors refer to DL19, but I think it would help readers evaluate this paper more quickly if a paragraph or so was taken to summarize DL19.

Pg. 5 ln 147: The authors refer to partitioning into precipitating and non-precipitating clouds. I am not clear how this is done. On line 245 it looks like 2C-RAIN-PROFILE is being used- this needs to be caveated that it will only see relatively heavy precipitation, but will miss light rain events. Other parts of the methodology makes it sound like AMSR-E precipitation is being used, which is problematic due to the AMSR-E precipitation just being a partitioning of condensed liquid by SST.

Pg. 6 Eq3-6: how do the authors account for CF being bounded between 0-1 in this calculation?

Pg. 7 Ln 197: The authors assert that they have accounted for the effects of precipitation on the aerosol-cloud-precipitation system. This is not supported by any evidence or citations, but is an important justification of the validity of the analysis presented here.

Pg. 8 ln 221: The authors assert that by binning LWP they reduce the chances of buffering. One thing that should be mentioned in this study is that AI and LWP will naturally anti-correlate due to precipitation and scavenging correlating with cloudiness (eg LWP or CF) (Wood et al., 2012) and due to air mass history leading to both drier

and more aerosol-laden air (Gryspeerdt et al., 2019). These non-causal relationships are not meaningful to ERFaci, but can substantially affect the covariability of cloud macrophysical properties and aerosol, and thus the inferred aci strength (McCoy et al., 2019). It is possible that the LWP binning and precipitation stratification reduce this effect. However, the authors must show some demonstration of the predictive ability of this method by either (1) applying it to GCM data (in this case SPRINTARS) and showing that their methodology when applied in a GCM can accurately reproduce the GCM response to enhance aerosol as in Gryspeerdt et al. (2016) or McCoy et al. (2019) – or – (2) examining one of the transient aerosol emissions identified in recent studies (Malavelle et al., 2017; Toll et al., 2019) and see if their characterization of sensitivity of cloud to aerosol has some predictive ability. Without this sort of test there is no guarantee that the inferred ERFaci_warm-topped_oceanic is accurate.

Pg 9 Ln 261: The authors find an ERFaci that at the very weakest end of what would be expected based on existing best-estimates (Bellouin et al., 2019). This is of course completely fine, but it would be interesting for the authors to add some discussion of why their forcing is so relatively weak compared to other empirical estimates of ERFaci from observations. The authors do cite the AR5 range, but this is for the range of GCM estimates, which may not be as appropriate to consider their results relative to as observational constraint studies. I suspect that this is partially because the authors are not really looking at ERFaci, but ERFaci_warm-topped_oceanic. As such I recommend the authors do not use the terminology ERFaci. In the interest of relating this result to forcing the authors could consider using this methodology applied to GCMs (as noted above I view this as a necessary condition to this analysis) and then using the GCMs to extrapolate this result to ERFaci as in Costa-Surós et al. (2019).

Pg 10 ln 300: An alternative explanation of the weakening precipitation effect in clouds with higher LWP may be that precipitation increases with LWP, which means that precipitation scavenging becomes larger, which in turn means that the true adjustment strength is obscured by non-causal covariance between aerosol and cloud macrophysics (see discussion in McCoy et al. (2019)).

Figure 7 and ln 456: The authors find a large ERFaci in the SH, which is really surprising given the very small change in anthropogenic aerosol in these regions. Figure 1 shows change in AI, but it is a bit hard to distinguish small changes from zero and the authors may want to consider some sort of log normalization to their color scale. However, strong ERFaci exists along a line around $40°$S, which is hard to square with studies examining pristine days in the PD (Hamilton et al., 2014). That is to say, the pattern of ERFaci in this study is dramatically different than the RFari shown in, for example, aerocom (Myhre et al., 2013).

Figure 7: While I think it's good to pursue analysis to its conclusion by applying it to all data, I am surprised at the positive RFaci and CA in the tropics. Can the authors comment on whether their analysis is sensitive to retrieval errors in convective cloud? In particular, a positive forcing due to RFaci is quite unusual- while it may be due to biomass burning aerosol above cloud in some regions via semi-direct effects or blocking reflective light (so not really aci) (Bellouin et al., 2019), the appearance of a positive RFaci seems to be more related to SST, than aerosol type given its appearance over the tropics, and far away from strong aerosol sources.

[Figure]

Review of: Quantifying Cloud Adjustments and the Radiative Forcing due to Aerosol-Cloud Interactions in Satellite Observations of Warm Marine Clouds By Douglas and L'Ecuyer

This study uses satellite observations with the addition of model aerosol data and re-analysis meteorological data to calculate the effective radiative forcing due to cloud-aerosol interaction in warm clouds over the oceans. The authors decompose the forcing to two components: due to the Twomey effect (RFaci), and due to cloud adjustments (CA), which in this case include only changes in cloud cover (without including changes

in LWP). The analysis is conducted also regionally and as a function of LWP, inversion strength and RH in the free troposphere. The binning according to the last two criteria is done to account for the meteorological dependency. The calculation is also done separately for precipitating and non-precipitating clouds. I think that this paper presents some interesting results that worth being published. However, I think that the paper includes some limitations that are not all fully acknowledged in the manuscript. Hence, including a more comprehensive discussion about these limitations and maybe weakening the conclusions accordingly will improve the paper.

General comments

1) If I understand correctly, calculating the radiative forcing based on the multiplication of the susceptibility calculated in present day with the total change in AI between present day and preindustrial assume linearity of the susceptibility with time. As you show that the susceptibility is a function of the environmental conditions and it is known that the environmental conditions changed, it is not clear how valid is this assumption. In addition, I think that your calculation assumes that the frequency of occurrence of each bin of EIS, RH and LWP remain the same between PD and PI (as you do not account for changes in the frequency of occurrence -eq. 9). I can't see any reasons for that to be true. Hence, and because of the large uncertainty in PI aerosol conditions, it might be better to stay only with the susceptibility calculations and not present the forcing calculations. I leave it to the authors to decide. 2) Feedbacks between clouds and the environmental conditions are not discussed and accounted for sufficiently. It is known that the environmental conditions may change differently under different aerosol conditions. In particular, the EIS and RH (maybe not at 700mb but definitely below that) may be affected by the clouds feedback on the environmental conditions differently under different aerosol conditions. In addition, direct aerosol-radiation interaction may influence the environmental conditions. Hence, the binning according to the meteorological conditions may not be independent of the aerosol conditions. I suggest to add a discussion about that. In addition, the separation to precipitating/non-precipitating

conditions could be, under certain conditions, due to aerosol effect (total rain suppression could be found in shallow clouds under polluted conditions). This effect is not discussed and you treat it as if it was external. 3) Co-variability between aerosol and cloudiness and the uncertainty in the causality relationships are not discussed enough. I appreciate that binning the data according to EIS and RH at 700 mb may reduce the role of co-variability between aerosol and cloudiness. However, some co-variability may still remain. For example, it was previously shown (Nishant and Sherwood, 2017) that under some conditions, near surface wind speed have a positive correlation with both aerosol concentration and cloudiness (CF in this case). It is possible, and even expected, that wind speed will be partially corelative with EIS and RH but not sure to what extent. I suggest to add a dissection about those limitations. 4) Uncertainties due to the semi-direct effect are not mentioned. Form satellite observations it is impossible to distinguish between the aerosol microphysical effect and the semi-direct effect but the latter is very likely to affect your calculations. I suggest to include a discussion about that. 5) Referring to the forcing only from warm marine cloud as ERFaci and RFaci might be confusing with the total estimations for all cloud. I appreciate that you mention the focus on warm clouds over the ocean directly in the tile and in many other places but I still think that the use of general terms here could be confusing. 6) At many places along the manuscript you mention "buffering" as if it was an artefact that one should avoid in his analysis (i.e. "While LWP being held approximately constant accounts for some variability in the meteorology, explicitly holding the stability and free atmospheric contributions fixed within regimes of EIS and RH700 will further control buffering and modulation of $\lambda$ by the environment."). I think that if indeed clouds under different aerosol levels change differently the environmental conditions to reduce the total aerosol effect, that is something important to understand. In addition, I think you don't properly define what you mean by "buffering". That term could be used to describe many mechanisms.

Specific comments:

L3: CA is not defined hear. Consider writing in full cloud adjustments.

L10: if RFaci and CA counteract and the total effect is small I would say that it could be attributed to damped susceptibility (or buffering). Why is it "erroneously"?

L21: what do you mean by "cloud forcing"? is it the cloud radiative effect? I think it is better not to use forcing here and stick with the common definition of radiative forcing.

L68-72: consider adding here that the sign of the effect was also shown to be a function of the background aerosol concentration.

L82: the non-monotonic response was shown for other cloud properties (such as cloud fraction and top height) as well as for precipitation. Hence, I don't understand why is it important to separate specifically this effect from the rest.

L105, L109, L141 and other places: again, maybe better to use radiative effect here instead of forcing.

L119: SPRINTARS was run (in the paper you are citing) in a T21 resolution ($\sim$5.6o) and hence is not "cloud resolving" at all.

L250-258: I couldn't really understand how the uncertainty was calculated. I think more details are needed for it to be reproducible. What is the magnitude of error added to PI and PD AI estimations? How did you choose this magnitude?

L265: you are comparing here the estimated forcing for only warm cloud over the ocean with the total estimation from the IPCC report. I don't think this competition is valid. In addition, in the introduction you cited a few papers showing that most of the ERFaci is coming from warm clouds over the ocean. How that can go together with the relatively low estimations you are getting for warm cloud compared to the total forcing?

L310: you cite here a paper focusing on deep convective clouds. Consider adding papers discussing warm cloud invigoration.

L 312: I don't understand the claim here. Why determining the casualty of aerosol

effect on LWP is more difficult than for CF?

L320: why is that a sign of "buffering"? it just means that the aerosol effect is non-monotonic and change sign. The aerosol level at which the sign flip is a function of the environmental condition as was shown before.

L417: the possible change in precipitation could also be relevant between PI and PD making point 1 (general comments above) even more critical.

L426-430: you don't mention here, at the beginning of the conclusion section, that these estimations are only relevant for warm cloud over the oceans. It could look like you are giving general estimations here.

L442: I think that this could also be due to the semi-direct effect of absorbing aerosols.

L445: again, if RFaci and CA counteract and the total effect is small I would say that it could be refer to as "buffering".

Technical comments L102 and L107: ECWMF -> ECMWF? Anyway, should be written in full (and maybe also add a citation).

L401: "on the both the"

Reference Nishant, N., and Sherwood, S. C.: A cloud‐resolving model study of aerosol‐cloud correlation in a pristine maritime environment, Geophysical Research Letters, 44, 5774-5781, 2017.

This study uses satellite observations with the addition of model aerosol data and reanalysis meteorological data to calculate the effective radiative forcing due to cloud aerosol interaction in warm clouds over the oceans. The authors decompose the forcing to two components: due to the Twomey effect (RFaci), and due to cloud adjustments (CA), which in this case include only changes in cloud cover (without including changes in LWP). The analysis is conducted also regionally and as a function of LWP, inversion strength and RH in the free troposphere. The binning according to the last two criteria is done to account for the meteorological dependency. The calculation is also done separately for precipitating and non precipitating clouds. I think that this paper presents some interesting results that worth being published. However, I think that the paper includes some limitations that are not all fully acknowledged in the manuscript. Hence, including a more comprehensive discussion about these limitations and maybe weakening the conclusions accordingly will improve the paper.

*We thank the reviewer for taking time to read our manuscript and provide constructive feedback. We will now go through and address each of their points.*

1) If I understand correctly, calculating the radiative forcing based on the multiplication of the susceptibility calculated in present day with the total change in AI between present day and preindustrial assume linearity of the susceptibility with time. As you show that the susceptibility is a function of the environmental conditions and it is known that the environmental conditions changed, it is not clear how valid is this assumption. In addition, I think that your calculation assumes that the frequency of occurrence of each bin of EIS, RH and LWP remain the same between PD and PI (as you do not account for changes in the frequency of occurrence -eq. 9). I can't see any reasons for that to be true. Hence, and because of the large uncertainty in PI aerosol conditions, it might be better to stay only with the susceptibility calculations and not present the forcing calculations. I leave it to the authors to decide.

*We agree that the frequency of occurrence for each regime changes throughout time, however we believe that it is useful to provide an estimate of the ERFaci for warm clouds under the assumption that the cloud and environmental regimes have not changed enough to significantly alter the distribution of states across our relatively coarse LWP, EIS, and RH bins.* A bin resolution of 10% in EIS, RH, and LWP is adopted to distinguish regimes in the present study. Even if these parameters have changed since pre-industrial times, it is unlikely that the changes are of a magnitude comparable to this bin resolution which would be required to substantially alter the distribution of states. *To place aerosol-cloud interaction effects into the context of other climate forcings, we feel it is important estimate how these susceptibilities, regime constraints, and regional variations combine to impact the overall estimated forcing. Never-the-less we focus on the sensitivities throughout the paper, with a majority of the results focused on how these change as a constraint is enforced. Future*

*work will include evaluating if climate models accurately capture current regime trends and examining how regimes may have changed since the pre-industrial times.*

2) Feedbacks between clouds and the environmental conditions are not discussed and accounted for sufficiently. It is known that the environmental conditions may change differently under different aerosol conditions. In particular, the EIS and RH (maybe not at 700mb but definitely below that) may be affected by the clouds feedback on the environmental conditions differently under different aerosol conditions. In addition, direct aerosol-radiation interaction may influence the environmental conditions. Hence, the binning according to the meteorological conditions may not be independent of the aerosol conditions. I suggest to add a discussion about that. In addition, the separation to precipitating/non-precipitating conditions could be, under certain conditions, due to aerosol effect (total rain suppression could be found in shallow clouds under polluted conditions). This effect is not discussed and you treat it as if it was external.

*Feedbacks between the clouds, the environment, and the aerosol are much harder to constrain as these are non-linear, time dependent, and occur on multiple time scales. While we agree that aerosol may alter some environmental conditions, the degree to which aerosol impacts the environment is much less than the degree to which aerosol affects clouds and/or clouds affect the environment. The free atmosphere should be minimally impacted by clouds or aerosol, except for in the highest humidity regime, where deep convection detrainment may lead to local changes in the $RH_{700}$. We agree that aerosol-radiation interactions can alter the environment. We have added some caveats to the Methodology and Observations 2.3 Regimes:*

> *"While binning our observations by environmental regime should control for some modulation the environment has on aerosol-cloud interactions, it does not fully capture aerosol-environment interactions. For example, in some regions such as off the coast of Africa, biomass burning results in smoke layers that absorb incoming radiation and warm the atmosphere (Cochrane, 2019). This could affect the humidity and temperature of the local environment. Environmental regime constraints would capture how the altered environment may regulate aerosol-cloud interactions, but separation into such regimes does not address how the aerosol has impacted the environment."*

3) Co-variability between aerosol and cloudiness and the uncertainty in the causality relationships are not discussed enough. I appreciate that binning the data according to EIS and RH at 700 mb may reduce the role of co-variability between aerosol and cloudiness. However, some co-variability may still remain. For example, it was previously shown (Nishant and Sherwood, 2017) that under some conditions, near surface wind speed have a positive correlation with both aerosol concentration and cloudiness (CF in this case). It is possible, and even expected, that wind speed will be partially correlative with EIS and RH but not sure to what extent. I suggest to add a dissection about those limitations.

*We agree that not all covariability will be limited by our regime constraints as they are do not encompass all meteorological variability of the boundary layer nor control*

*for all processes that may impact clouds, aerosols, or their interactions. We have added to Methodology and Observations 2.3 Regimes:*

> *"Using EIS and $RH_{700}$ does not guarantee to limit all covariability between the environment, aerosols, clouds, and their interactions. Some covariability may still exist, such as surface winds that may affect both clouds and aerosol (Nishant and Sherwood, 2017). These constraints only account for the major environmental controls on clouds and aerosol-cloud interactions, some more minor or less common environmental controls may still exert an influence on our results."*

*We have also added in Results and Discussion section 3.3 Constrained by local meteorology*

> *"It is possible with additional constraints, understanding how other components of the meteorology is affecting these terms would become more clear."*

4) Uncertainties due to the semi-direct effect are not mentioned. From satellite observations it is impossible to distinguish between the aerosol microphysical effect and the semi-direct effect but the latter is very likely to affect your calculations. I suggest to include a discussion about that.

*In Results and Discussion section 3.6 we do discuss how the semi-direct effect may be influencing our estimates of the RFaci. We have added, for further clarity, in Results and Discussion section 3.3. Constrained by local meteorology*

> *"It is also possible lambda_RFaci is impacted by some semi-direct effects by smoke aerosol which would lead to a cloud dimming and positive susceptibility. Semi-direct effects are not accounted for by our methodology, however aerosol within the cloud layer could lead to cloud breakup processes, a dimmer albedo, and changes to the local environment by the absorbing aerosol."*

5) Referring to the forcing only from warm marine cloud as ERFaci and RFaci might be confusing with the total estimations for all cloud. I appreciate that you mention the focus on warm clouds over the ocean directly in the tile and in many other places but I still think that the use of general terms here could be confusing.

*We agree with your suggestion. Sometimes it is easy to focus on warm clouds and forget other clouds are important to the climate too. We have changed ERFaci, RFaci, and CA to be $ERFaci_{warm}$, $RFaci_{warm}$, $CA_{warm}$ in order to remind the reader these results are for only one cloud type.*

6) At many places along the manuscript you mention "buffering" as if it was an artefact that one should avoid in his analysis (i.e. "While LWP being held approximately constant accounts for some variability in the meteorology, explicitly holding the stability and free atmospheric contributions fixed within regimes of EIS and RH700 will further control buffering and modulation of λ by the environment."). I think that if indeed clouds under different aerosol levels change differently the environmental conditions to reduce the total aerosol effect, that is something important to understand. In addition, I think you don't properly define what you mean by "buffering". That term could be used to describe many mechanisms.

*We have added to Methodology and Observations 2.3 Regimes:*
> *"Buffering can entail the cloud being too thick and impervious to changes due to aerosol due to its high LWP, offsetting and opposite reactions of the cloud resulting in reduced mean signal, or the environment acting to damp the cloud reaction, such as an unstable boundary layer reducing the impact of aerosol on cloud lifetime (Fan, 2016; Stevens & Feingold 2007)."*

*We have also removed some references of buffering to simplify some explanations.*
*We have added to Results and Discussion 3.2*
> *"Modulation may by the environment can include the amplification of the reaction through a stable environment further prolonging the cloud lifetime and therefore extent."*

*In general, there are so many different environmental or liquid water path dependent processes that could affect aerosol-cloud interactions that to go through those all would be a review paper in itself.*

Specific Comments
L3: CA is not defined hear. Consider writing in full cloud adjustments.
*Added cloud adjustments before CA is used.*
L10: if RFaci and CA counteract and the total effect is small I would say that it could be attributed to damped susceptibility (or buffering). Why is it "erroneously"?
*We have removed buffering so as not to confused the reader in this aspect. The total susceptibility may be small, however that does not mean the individual components are each small or cause cooling. A point we aim to make is the idea that the ERFaci should be considered by its components in order to better understand all processes occurring.*
L21: what do you mean by "cloud forcing"? is it the cloud radiative effect? I think it is better not to use forcing here and stick with the common definition of radiative forcing.
*We have replaced cloud forcing with cloud radiative effect.*
L68-72: consider adding here that the sign of the effect was also shown to be a function of the background aerosol concentration.
*We have added "and the background state of the aerosol."*
L82: the non-monotonic response was shown for other cloud properties (such as cloud fraction and top height) as well as for precipitation. Hence, I don't understand why is it important to separate specifically this effect from the rest.
*We have added "in order to reduce the effects of this non-linear relationship on our results."*
L105, L109, L141 and other places: again, maybe better to use radiative effect here instead of forcing.
*We have replaced some instances of forcing with flux.*
L119: SPRINTARS was run (in the paper you are citing) in a T21 resolution (~5.6o) and hence is not "cloud resolving" at all.
*We have removed cloud resolving.*
L250-258: I couldn't really understand how the uncertainty was calculated. I think more details are needed for it to be reproducible.

*We have added: The regressions within all regime constraints, from only meteorological to regional, remain robust for all susceptibilities when 10% of the AI estimates were randomly assigned.*

What is the magnitude of error added to PI and PD AI estimations? How did you choose this magnitude?

*The PI aerosol error magnitude is calculated from the SPRINTARS data.*

L265: you are comparing here the estimated forcing for only warm cloud over the ocean with the total estimation from the IPCC report. I don't think this competition is valid. In addition, in the introduction you cited a few papers showing that most of the ERFaci is coming from warm clouds over the ocean. How that can go together with the relatively low estimations you are getting for warm cloud compared to the total forcing?

*Our estimates remain at the low end of observation-based estimates of ERFaci for warm clouds. IPCC estimates are primarily based on global climate models, which difference industrial vs. non-industrial runs.*

*It is possible that even these estimates of forcing are slightly different than the definition of forcing from the IPCC or model based studies which difference top-of-atmosphere forcings in polluted vs. non-polluted GCM runs.*

*In the methods section. Our methodology agrees with how others have calculated ERFaci from observations, however.*

L310: you cite here a paper focusing on deep convective clouds. Consider adding papers discussing warm cloud invigoration.

*We have added a citation to a warm cloud invigoration paper by Ilan Koren.*

L 312: I don't understand the claim here. Why determining the casualty of aerosol effect on LWP is more difficult than for CF?

*Research has yet to agree if there is some effect of aerosol on LWP (Toll et al. 2019), a large effect (Rosenfeld et al. 2019), or non-linear effects. As deriving a signal in liquid water susceptibility has proven difficult, we chose to focus only on cloud extent, where research has converged to more of an agreement.*

L320: why is that a sign of "buffering"? it just means that the aerosol effect is nonmonotonic and change sign. The aerosol level at which the sign flip is a function of the environmental condition as was shown before.

*We have replaced "buffering effect" with "the influence of the environment"*

L417: the possible change in precipitation could also be relevant between PI and PD making point 1 (general comments above) even more critical.

*We agree that precipitation, its effects on aerosol and the environment, and how this alters aerosol-cloud interactions is important. Since we addressed your first general comment, we believe this will now tie in well with our final statement in the precipitation section.*

L426-430: you don't mention here, at the beginning of the conclusion section, that these estimations are only relevant for warm cloud over the oceans. It could look like you are giving general estimations here.

*We have added "warm, marine cloud" before ERFaci in this first sentence. This, along with adding the warm subscript, should remind the reader these results are limited to only warm clouds.*

Technical comments L102 and L107: ECWMF -> ECMWF? Anyway, should be written in full (and maybe also add a citation). *Fixed.*
L401: "on the both the"
*Fixed to "on both the."*

Reviewer 1 Response
* * *
The authors utilize remote sensing observations and a regime-based approach to isolate the effects of varying aerosol index on cloud microphysical (1st indirect effect) and cloud macrophysical properties (adjustments). The authors utilize regimes of above cloud RH and stability. LWP is binned to account for variations in cloud state in each regime. The results show that in some regions adjustments and the first indirect effect have opposing signs. The authors also show that as LWP increases the radiative response to AI saturates. The analysis presented here satisfies the important problem of separating variability due to meteorology from aerosol-cloud interactions (aci). The authors find a relatively weak ERFaci from warm-topped clouds over oceans, which appears to be due to dimming in regions in the equatorial Atlantic and Indian ocean.

*We would like to thank the reviewer for taking the time to read our manuscript and provide feedback and comments.*

While I appreciate that the authors are applying the methodology developed in a previous study, it is hard to understand what is being done and I think the authors could briefly summarize their methodology to allow readers to more efficiently refer to DL19. The description of the observational data sets could be much more substantial. It is confusing what observational and modeling data is being used for what. In some cases it appears that observational data sets that are not appropriate are being used, but it is hard to confirm this from the data section. One solution that might make this un-ambiguous would be to create a table of variables and data sources.

*We are only using satellite observations and reanalysis data intended to be paired with satellite observations (MERRA-2). To clarify what observations we are using, we have added to section 2.1 Data:*

> *"Collocated satellite observations of cloud shortwave forcing, cloud fraction, and aerosol index are obtained by NASA A-Train satellites Aqua, CloudSat, and The Cloud-Aerosol Lidar and Infrared Pathfinder Satellite Observation (CALIPSO) from 2007 to 2010. The NASA A-Train is configured to maximize the synergy between different satellite products to improve our understanding of clouds, aerosols, and the environment (L'Ecuyer et al. 2011)."*
> *"2B-CLDCLASS-LIDAR combines CloudSat's CPR with CALIPSO lidar observations in order to discern even the thinnest clouds."*
> *"To broadly characterize large-scale environmental conditions, MERRA-2 temperature and humidity profiles are collocated by taking the environmental profile within 30 minutes of a CloudSat overpass and within ~1/2 degree latitude and longitude"*

A critical issue with this paper is use of area-mean LWP (in-cloud LWP*CF) from microwave when the authors imply they are using in-cloud LWP based on wording in the paper (ln 153). From reading the discussion in DL19 I believe that scene-mean LWP from AMSR is just used to filter data into rough bins, and does not play a role in the analysis beyond this. While this is probably not a big problem, the authors may want to clarify what the footprints of the different data sets are that they are using, possibly with a diagram overlaid over an actual satellite image to allow readers who are less familiar with remote sensing to contextualize what is being shown, especially because the authors are using active instruments averaged along track with passive instruments. In particular, in this regard I am confused how the authors are overlapping along-track averaged CF from Cloudsat-CALIPSO with AMSR LWP and a diagram might be helpful. A nice image of the actual cloud field from MODIS on the background would be helpful to readers trying to contextualize the retrievals in terms of cloud features.

*We have added the caveats of the footprint discrepancies along with how close geometrically the footprints are. Added to section 2.1 Data:*
> *"While the footprints of CloudSat and AMSR-E do not perfectly overlap, the AMSR-E LWP is used to establish a scene based constraint on the clouds in order to better consolidate our observations into regimes. AMSR-E's footprint is within ~2.5 km of CloudSat's track, meaning both sensors are observing the same, liquid clouds (Lebsock et al. 2014)."*

*CloudSat observations are often combined with AMSR-E scene averaged LWP in a number of cloud and aerosol studies (such as L'Ecuyer et al. 2009 and Chen et al. 2014). Our study does not aim to understand how the LWP responds to aerosol, only to use LWP as a higher level constraint in order to partition warm clouds into characteristic regimes.*

The authors need to either apply their analysis in a GCM simulating PI and PD (Gryspeerdt et al., 2017; Gryspeerdt et al., 2016; McCoy et al., 2019; Costa-Surós et al., 2019) to make sure that their analysis methodology has predictive power, or examine the response of cloud to some sort of transient change in aerosol (Malavelle et al., 2017; Toll et al., 2019) and make sure that their analysis trained over different data can predict the response to the transient change in AI. Without these falsification tests of their predictions, it is unclear what predictive use their correlation model has in that there is no way to falsify their predictions. Even an approximate calculation using model LWP, CF, SW, and AI without any complex output along the satellite overpass (which doesn't appear to be a major source of error compared to problems from low aerosol amount as shown in Ma et al. (2018)) would provide a much more powerful validation of what the authors are hypothesizing is the ERFaci.

*A next step will be to find these same signals within output from a GCM, however that is beyond the scope of the current study. This study intends to only document how the observed brightness and extent of clouds respond as aerosol concentration increases, and how these signals depend on the environment and cloud state.  These responses*

*are then used to derive an estimate of ERFaci that is consistent with the specific observations used. While similar methods can be applied to GCM output, the results do not provide a stringent test on the methodology since model responses will depend strongly on how the underlying processes are represented in the model. Non-linearities or stronger/weaker dependencies on environmental state may yield vastly different results that do not provide a useful assessment of the validity of the decomposition approach. Furthermore, any meaningful comparison of GCM output against observations is severely limited by mismatches in resolution between the large GCM gridbox and the fine-scale satellite observations (e.g. Kay et al, 2019).*

*In principle, results from this study can be used to assess how well GCMs recreate the derived linearized relationships between aerosol, cloud brightness, and cloud extent under different environmental regimes but such an evaluation requires considerable additional effort and requires close cooperation with modeling groups to ensure appropriate interpretation of the results. It is acknowledged in the manuscript that our study merely aims to document the observed relationships in present climate, not to predict how these may have changed since pre-industrial conditions. Our study provides a benchmark of regimes to be used to evaluate how well updated parameterizations capture current signals.*

*Within section 2.4 Decomposing the ERFaci we point out that we do not use the lowest 12% of aerosol indices in order to reduce biases in regimes where the correlation between our aerosol proxy and CCN is expected to be weak.*

The authors ultimately present a correlative study to predict ERFaci (or at least ERFaci for warm-topped clouds over oceans- see comments below). Characterizing covariance is important but does not guarantee an accurate prediction. In the case of aerosol-cloud adjustments in particular, there is not a unique causality flowing from aerosol to cloud (Wood et al., 2012; Gryspeerdt et al., 2019). In this context, and because their ERFaci is rather weak compared to other studies it seems possible that their analysis conflates aci with precipitation scavenging and other confounders (Gryspeerdt et al., 2019), which would tend reduce correlation strength between aerosol and cloud amount (eg precipitation scavenging is strongest when there is a lot of cloud and there tend to be less cloud and more aerosol off the coast of continents).

*It should first be noted that our estimate of the warm cloud ERFaci is within the limits of uncertainty (±0.16 Wm⁻²) of other observation based estimates such as Christensen et al. (-0.36 Wm⁻²). To address potential biases due to scavenging effects, we explicitly control for precipitation using CloudSat observations that represent the most sensitive satellite-based metric for precipitation occurrence (Haynes et al, 2009). Separating precipitating from non-precipitating clouds in order to understand how precipitation scavenging and other processes that differ between the two alter their ERFaci reduces our decomposed estimate from -0.21 to -0.207 Wm⁻² . If our estimates were highly affected by precipitation scavenging of aerosol, we would expect the difference between these estimates to be greater.*

*We acknowledge that our regimes do not capture all signals of covariability between the environment and aerosol and have added to section 2.2 Regimes:*
*"Using EIS and $RH_{700}$ does not guarantee to limit all covariability between the environment, aerosols, clouds, and their interactions. Some covariability may still exist, such as surface winds affecting both clouds and aerosol (Nishant et al. 2017)."*

The authors need to refer to their ERFaci as ERFaci_liquid-topped_over_oceans (or at least that is my take from the methodology and Eq 9).

*We have changed all mentions of ERFaci to $ERFaci_{warm}$, RFaci has become $RFaci_{warm}$, and CA has become $CA_{warm}$ in order to remind the reader these results only apply for warm-topped clouds. We have added mentions of our observations being limited to only marine warm clouds throughout section 2.1 Data.*

Specific changes:
Pg 1 ln 3: ERFaci is a combination of microphysical (RFaci) and macrophysical changes (adjustments) and the latter could be further split into changes in extent and thickness (Gryspeerdt et al., 2019; Gryspeerdt et al., 2017; Gryspeerdt et al., 2016). As written this implies that thickness stays constant and the only possible adjustment is CF. I understand now that this is more like the intrinsic extrinsic separation in other studies (Christensen et al., 2017), but this would be better to clarify in the abstract.

*We have added to the abstract intrinsic and extrinsic to make the connection to the study by Chen et al. 2014 and Christensen et al. 2017 adding next to the RFaci term intrinsic and the cloud adjustment term extrinsic.*

Pg. 2 ln 40: The goals of DL19 overlap a lot with the goals of the present study. A sentence like 'The present study expand on DL19 in the following ways:' would be helpful. I believe the primary difference between these studies is the inclusion of adjustments, but it would be helpful to state that explicitly for readers to rapidly ingest what is happening.
*We have added to section 1 Introduction:*
> *"The present study expands upon work done in DL19 by specifying what aspects of the cloud lead to changes in the CRE, whether that be the brightness or cloud extent or both, and whether these changes can negate each other, such as when a cloud shrinks but the brightness increases."*

Pg. 3 ln 85: It would help readers to quickly process what data sets are being used to describe what variable to use subheaders here (2.1 Data, 2.1.1 Warm cloud fraction). This is stylistic, but I found it hard to understand where precipitation measurements were coming from. I think that it would help a lot to have a table of what the precise data sets used are, especially since some of the remote sensing data

sets being used may be inappropriate, but it is unclear if they are actually used (eg AMSR rain rates, although I believe these are not used despite being mentioned).

We have added an additional paragraph in section 2.1 Data to clarify how we separated precipitating and non-precipitating clouds exactly.

> *"Clouds are separated into precipitating and non-precipitating regimes using CloudSat's 2C-PRECIP-COLUMN precipitation flag. Clouds with a 0 precipitation flag, no precipitation detected, are designated as non-precipitating. Precipitating clouds are separated using flag 3, where rain is certain (Haynes et al. 2009). Our precipitating clouds include a majority of the drizzling cases, as CloudSat's 2C-PRECIP-COLUMN's threshold for drizzle is -15 dB, which should capture all but the lightest drizzling clouds (Stephens et al. 2007)."*

Pg4 ln 124: is the material not shown in the citation? If it's in the citation no need to put not shown here.
*The material is shown within the citation, we meant to say that we do not show these results within the current paper. We have removed not shown.*

Pg 4 ln 125: Swelling is a key issue in trying to understand adjustments. I believe that swelling is not an issue for SPRINTARS because the model can be internally consistent, but an additional comment is needed about MACC aerosol swelling. It's unclear that MACC can fully correct for swelling given the very complex way that swelling occurs (Christensen et al., 2017; Twohy et al., 2009). This needs to be explained and caveated. Also, why mix MACC aerosol and MERRA-2 meteorology? MERRA2 produces a very similar aerosol reanalysis to MACC and this would avoid confusing MERRA2 meteorology with aerosol reanalysis in a different framework. Also- how are SPRINTARS and MACC not sensitive to precipitation scavenging? Presumably both data sets have a precipitation sink of aerosol otherwise it would be very hard to maintain realistic aerosol.

*Our results shown do not include any MACC aerosol products. We removed the reference to MACC aerosol in order to not confuse the reader. We have done the same regime analysis with MACC and SPRINTARS AOD for the same time period in order to validate the sign of the regime signals derived here. We have removed the precipitation scavenging mention since SPRINTARS does include some type of precipitation sink for aerosol.*

*We have added to section 2.4 Decomposing the ERFaci:*
*"Aerosols swell in the vicinity of clouds, which increases their size and therefore affects the MODIS retrieval AI (Christensen et al. 2017). To assess how significantly this may affect results we have randomly added errors of 10% to our AI estimates and re-derived all signals with all regime constraints. Even with extreme amounts of error in AI, the signals within our environmental and LWP regimes are robust.*

Pg.5 Ln140: This methods section is really short. I understand that the authors refer to DL19, but I think it would help readers evaluate this paper more quickly if a paragraph or so was taken to summarize DL19.

*We have added to the methods:*
> *"In DL19, environmental and cloud state regimes were imposed on a regional basis in order to identify regime specific behavior of aerosol-cloud-radiation interactions. Within each regime, we regressed the cloud radiative effect (CRE) against AI in order to find the susceptibility of warm cloud radiative properties to aerosol. We use these same susceptibilities within section 3.1 to quantify the total warm, marine ERFaci. DL19 found that the susceptibility varies regionally and by regime, however the ERFaci$_{warm}$ depends on the magnitude to which aerosol has increased since pre-industrial times. Further, the ERFaci$_{warm}$ does not diagnose what characteristics of the cloud are causing the effect, prompting us within this paper to decompose the ERFaci$_{warm}$ into the effects on the albedo and the effects on cloud extent."*

Pg. 6 Eq3-6: how do the authors account for CF being bounded between 0-1 in this calculation?
*Our cloud fraction is the fraction of a 12 km x 1 km along track region covered in clouds according to CloudSat's 2B-CLDCLASS-LIDAR, which includes even the thinnest clouds not captured by CPR. Therefore, our cloud fractions should be between 0 and 1.*

Pg. 8 ln 221: The authors assert that by binning LWP they reduce the chances of buffering. One thing that should be mentioned in this study is that AI and LWP will naturally anti-correlate due to precipitation and scavenging correlating with cloudiness (eg LWP or CF) (Wood et al., 2012) and due to air mass history leading to both drier and more aerosol-laden air (Gryspeerdt et al., 2019). These non-causal relationships are not meaningful to ERFaci, but can substantially affect the covariability of cloud macrophysical properties and aerosol, and thus the inferred aci strength (McCoy et al., 2019). It is possible that the LWP binning and precipitation stratification reduce this effect. However, the authors must show some demonstration of the predictive ability of this method by either (1) applying it to GCM data (in this case SPRINTARS) and showing that their methodology when applied in a GCM can accurately reproduce the GCM response to enhance aerosol as in Gryspeerdt et al. (2016) or McCoy et al. (2019) – or – (2) examining one of the transient aerosol emissions identified in recent studies (Malavelle et al., 2017; Toll et al., 2019) and see if their characterization of sensitivity of cloud to aerosol has some predictive ability. Without this sort of test there is no guarantee that the inferred ERFaci_warm-topped_oceanic is accurate.

*We have added to section 3.2 Impact of LWP within the results:*
> *"While regime constraints on LWP do reduce the covariability between aerosol-cloud interactions and the role LWP plays in buffering these interactions, it does not remove all sources of covariability between LWP, aerosol, the*

*environment, and cloud properties. Aerosol has been shown to negatively correlate with LWP (Gryspeerdt et al. 2019). It is possible that this relationship, and the inherent relationship between the environment and LWP, could affect results shown."*

*Future work is planned to evaluate how regime-specific relationships compare to those derived via application of similar methods to GCMs, however, as noted above, uncertainty in the parameterization of aerosol-cloud interactions and their regime-dependence preclude drawing concrete conclusions regarding the validity of the methodology from such analyses.  More importantly, the resolution of today's GCMs is not sufficient to accurately emulate the distributions of clouds and aerosols on the same scales as the observations so considerable thought and effort will be needed to ensure that the methods can be applied within a model framework in a meaningful way. We agree that the observations have caveats, which we have acknowledged within our manuscript, but we have thoroughly documented our methods, the underlying datasets used, and the analysis approach.  As with any study, these choices can be debated and improved upon but the results presented here are (a) an accurate representation of the correlations that exist in the datasets employed; (b) reproduceable; and (c) accompanied by an appropriately large error bar.  We believe these data and the analysis method described here represent the current state of the art given current Earth observing capabilities but acknowledge that these estimates will likely be refined in the future.*

Pg 10 ln 300: An alternative explanation of the weakening precipitation effect in clouds with higher LWP may be that precipitation increases with LWP, which means that precipitation scavenging becomes larger, which in turn means that the true adjustment strength is obscured by non-causal covariance between aerosol and cloud macro physics (see discussion in McCoy et al. (2019)).

*We have added to section 3.2 Impact of LWP:*
> *"An alternative explanation is that thicker clouds with larger LWPs are more likely to precipitate, scavenging aerosol and weakening the susceptibility. Aerosol-cloud-precipitation interactions complicate cloud adjustment processes in higher LWP clouds; the true susceptibility may be masked by covariance between aerosol and precipitation in these clouds (McCoy et al. 2019)."*

Figure 7 and ln 456: The authors find a large ERFaci in the SH, which is really surprising given the very small change in anthropogenic aerosol in these regions. Figure 1 shows change in AI, but it is a bit hard to distinguish small changes from zero and the authors may want to consider some sort of log normalization to their color scale. However, strong ERFaci exists along a line around 40◦S, which is hard to square with studies examining pristine days in the PD (Hamilton et al., 2014). That is to say, the pattern of ERFaci in this study is dramatically different than the RFari shown in, for example, aerocom (Myhre et al., 2013).

*Since our estimates of the ERFaci are weighted by occurrence, regions with the highest occurrence of warm clouds will have larger ERFaci. The southern hemisphere is known to have the largest occurrence of warm cloud decks, therefore our weighted ERFaci from observations will weight the southern hemisphere over the northern hemisphere. Further, the southern ocean may have a higher susceptibility due to their usual pristine conditions making them primed and highly sensitive to any changes in aerosol.*

Figure 7: While I think it's good to pursue analysis to its conclusion by applying it to all data, I am surprised at the positive RFaci and CA in the tropics. Can the authors comment on whether their analysis is sensitive to retrieval errors in convective cloud? In particular, a positive forcing due to RFaci is quite unusual- while it may be due to biomass burning aerosol above cloud in some regions via semi-direct effects or blocking reflective light (so not really aci) (Bellouin et al., 2019), the appearance of a positive RFaci seems to be more related to SST, than aerosol type given its appearance over the tropics, and far away from strong aerosol sources.

*A limitation of our data is that the cloud radiative effect can be reduced due to semi-direct effects not constrained by our environmental or LWP limits.*
*We have added to address that the semi-direct effect is not accounted for by our methodology and may result in a reduced RFaci, in Results and Discussion section 3.3. Constrained by local meteorology*
> *"It is also possible lambda_RFaci is impacted by some semi-direct effects by smoke aerosol which would lead to a cloud dimming and positive susceptibility. Semi-direct effects are not accounted for by our methodology, however aerosol within the cloud layer could lead to cloud breakup processes, a dimmer albedo, and changes to the local environment by the absorbing aerosol."*

**Quantifying Cloud Adjustments and the Radiative Forcing due to Aerosol-Cloud Interactions in Satellite Observations of Warm Marine Clouds**

Alyson Douglas[1] and Tristan L'Ecuyer[1, 2]

[1]University of Wisconsin-Madison 1225 W. Dayton St Madison, WI
[2]Cooperative Instiue for Meteorological and Satellite Studies 1225 W. Dayton St Madison, WI

**Correspondence:** Alyson Douglas (ADouglas2@wisc.edu)

**Abstract.** Aerosol-cloud interactions and their resultant forcing remains one of the largest sources of uncertainty of future climate scenarios. The effective radiative forcing due to aerosol-cloud interactions (ERFaci) is a combination of two different effects, how aerosols modify cloud brightness (RFaci, intrinsic) and how cloud extent reacts to aerosol (cloud adjustments CA, extrinsic). Using satellite observations of warm clouds from the NASA A-Train constellation from 2007 to 2010 along with MERRA-2 reanalysis and aerosol from the SPRINTARS model, we evaluate the ERFaci [..[1] ]in warm, marine clouds and its components, the $RFaci_{warm}$ and $CA_{warm}$, while accounting for the liquid water path and local environment. We estimate the $ERFaci_{warm}$ to be -0.32 $\pm$0.16 Wm$^{-2}$. The $RFaci_{warm}$ dominates the $ERFaci_{warm}$ contributing 80% (-0.21 $\pm$0.15 Wm$^{-2}$), while the $CA_{warm}$ 
[revised manuscript text omitted]